# Decapping activators Edc3 and Scd6 act redundantly with Dhh1 in post-transcriptional repression of starvation-induced pathways

Rakesh Kumar[1†], Fan Zhang[1†], Shreyas Niphadkar[2], Chisom Onu[3], Anil Kumar Vijjamarri[1], Miriam L Greenberg[3], Sunil Laxman[2], Alan G Hinnebusch[1*]

[1]Division of Molecular and Cellular Biology, Eunice Kennedy Shriver National Institute of Child Health and Human Development, National Institutes of Health, Bethesda, United States; [2]Institute for Stem Cell Science and Regenerative Medicine (DBT-inStem), Bengaluru, India; [3]Department of Biological Sciences, Wayne State University, Detroit, United States

**\*For correspondence:**
ahinnebusch@nih.gov

[†]These authors contributed equally to this work

## eLife Assessment

This **important** study reports on the redundant roles of the decapping activators Edc3 and Scd6 in orchestrating post-transcriptional programs to modulate metabolic responses to nutrients in yeast. The authors employed mutagenesis studies in conjunction with a battery of transcriptome-wide analyses to provide **convincing** evidence supporting their conclusions. Considering the broad implications of post-transcriptional regulation of gene expression, this study will be of interest across a variety of biomedical disciplines ranging from biochemistry and molecular and cellular biology to those specializing in studying various pathologies.

**Abstract** Degradation of many yeast mRNAs involves decapping by the Dcp1:Dcp2 complex. Previous studies on decapping activators Edc3 and Scd6 suggested their limited roles in mRNA decay. RNA-seq analysis of mutants lacking one or both proteins revealed that Scd6 and Edc3 have largely redundant activities in targeting numerous mRNAs for degradation that are masked in the single mutants. These transcripts are frequently targeted by decapping activators Dhh1 and Pat1, and the collective evidence suggests that Scd6/Edc3 act interchangeably to recruit Dhh1 to Dcp2. Ribosome profiling shows that redundancy between Scd6 and Edc3 and their functional interactions with Dhh1 and Pat1 extend to translational repression of particular transcripts, including a cohort of poorly translated mRNAs displaying interdependent regulation by all four factors. Scd6/Edc3 also participate with Dhh1/Pat1 in post-transcriptional repression of proteins required for respiration and catabolism of alternative carbon sources, which are normally expressed only in limiting glucose. Simultaneously eliminating Scd6/Edc3 increases mitochondrial membrane potential and elevates metabolites of the tricarboxylic acid and glyoxylate cycles typically observed only during growth in low glucose. Thus, Scd6/Edc3 acts redundantly, in parallel with Dhh1 and in cooperation with Pat1, to adjust gene expression to nutrient availability by controlling mRNA decapping and decay.

## Introduction

Degradation of mRNA is a key aspect of gene expression that can be regulated in response to nutrient availability, cell stress, and developmental pathways in eukaryotic cells and further serves to eliminate defective mRNAs. A major pathway of cytoplasmic mRNA turnover involves truncating the poly(A) tail by the Ccr4/Not and Pan2/Pan3 complexes, followed by removal of the $m^7G$ cap by the Dcp1/Dcp2 decapping complex and 5' to 3' exonucleolytic degradation by Xrn1. The decapping complex is activated by factors that interact with low-complexity sequence motifs in the C-terminal tail (CTT) of the catalytic subunit Dcp2, including Edc3, Scd6, DEAD-box helicase Dhh1, and Pat1, which also interact with one another extensively. There is evidence that Pat1 is recruited to oligoadenylate tails remaining on mRNAs following partial deadenylation, in association with the Lsm1-Lsm7 complex, and activates decapping via interactions with other decapping activators and with the Dcp2 CTT itself (*Parker, 2012*; *He and Jacobson, 2023*).

Genome-wide analysis of mRNA abundance (RNA-seq) in yeast mutants lacking Dhh1, Pat1, or both factors revealed functional cooperation by Pat1 and Dhh1 (*He et al., 2018*), with a large fraction being up-regulated only in the *pat1Δdhh1Δ* double mutant (*Vijjamarri et al., 2023a*). The majority of mRNAs up-regulated by *dhh1Δ* or *pat1Δ* mutations are likewise up-regulated in *dcp2Δ* cells (*He et al., 2018*) and exhibit greater than average proportions of decapped mRNAs in WT cells but not in *dhh1Δ* or *pat1Δ* cells (*Vijjamarri et al., 2023a*), consistent with Dhh1/Pat1 targeting mRNAs for degradation via decapping. While more than half of the mRNAs up-regulated by *dcp2Δ* are up-regulated by *dhh1Δ* and/or *pat1Δ*, a large fraction are targeted primarily by the Upf factors instead, responsible for nonsense-mediated mRNA decay (NMD) (*Vijjamarri et al., 2023b*; *Celik et al., 2017*).

The cumulative contributions of Dhh1 and Pat1 to mRNA decay is consistent with their independent interactions with distinct segments of the Dcp2 CTT (*He et al., 2018*), which appears to be direct for Pat1 but bridged by Edc3 or Scd6 for Dhh1. There is also evidence for distinct decapping complexes containing either Dhh1 or Pat1 in addition to Xrn1, Edc3, or Scd6 (*He et al., 2022*). However, Dhh1 occupancy (*Miller et al., 2018*), tends to be elevated for mRNAs up-regulated by either *dhh1Δ* (23) or *pat1Δ* (*Vijjamarri et al., 2023a*), consistent with Dhh1 contributing to degradation of mRNAs targeted by Pat1. Moreover, Pat1 binding to the Dcp2 CTT was required for degradation of certain Dhh1-targeted mRNAs when Dhh1 recruitment to the CTT was compromised (*He et al., 2022*).

There is evidence that Edc3 is a common constituent of decapping complexes containing Xrn1 and one or more of the decapping activators Dhh1, Pat1, Scd6, and Upf1 (*He et al., 2022*). It is surprising, therefore, that Edc3 has been implicated in targeting only two transcripts for degradation, *YRA1* and *RPS28B* (*Badis et al., 2004*; *Dong et al., 2007*). Edc3 shares sequence similarity with decapping activator Scd6 and both proteins contain an FDF motif shown to interact competitively with the Dhh1 homolog in animal systems (*Tritschler et al., 2008*; *Tritschler et al., 2009*). Edc3 and Scd6 also share N-terminal LSm domains that compete for binding to helical-leucine-rich (HLM) motifs in the CTT of fission yeast Dcp2 (*Fromm et al., 2012*). Analyzing effects of deleting the Edc3 interaction site in the *S. cerevisiae* Dcp2-CTT on levels of several Dhh1-repressed mRNAs suggested that recruitment of Dhh1 to Dcp2 can be mediated interchangeably by Edc3 or Scd6 bound to the same site in the Dcp2 CTT (*He et al., 2022*). These findings, plus the fact that deleting *S. cerevisiae SCD6* and *EDC3* simultaneously confers a synthetic growth defect (*Decourty et al., 2008*), suggest that yeast Scd6 and Edc3 function redundantly in targeting specific mRNAs for decapping and degradation.

Although tethering yeast Dhh1 or Scd6 enhances degradation of reporter mRNAs via Dcp1/Dcp2 (*Carroll et al., 2011*; *Sweet et al., 2012*; *Zeidan et al., 2018*), it is not well understood how these factors are targeted to specific native mRNAs. Dhh1 has been implicated in accelerating degradation associated with non-optimal codons in yeast mRNAs (*Presnyak et al., 2015*), being required for the rapid turnover conferred by suboptimal codons inserted in reporter mRNAs (*Sweet et al., 2012*). A queue of slowly elongating ribosomes upstream from non-optimal codons can be recognized by Dhh1, and overexpressing or tethering Dhh1 evokes ribosome stalling at non-optimal codons. Moreover, Dhh1 association and Dhh1-dependent repression of mRNA abundance both correlate with codon non-optimality across the yeast transcriptome (*Radhakrishnan et al., 2016*). However, the mRNAs most highly up-regulated in *dhh1Δ* and *pat1Δ* cells are not enriched for suboptimal codons, suggesting that other features are responsible for their preferential targeting by Dhh1 or Pat1 for decapping/decay (*Vijjamarri et al., 2023a*).

In addition to enhancing mRNA decay, there is evidence that Pat1, Dhh1, and Scd6 can repress translation. Tethering Dhh1 or Scd6 represses translation of reporter mRNAs in *dcp2Δ* cells, where the tethered transcripts cannot be decapped and degraded (*Carroll et al., 2011*; *Sweet et al., 2012*; *Zeidan et al., 2018*). Deletion of Dhh1 and Pat1 simultaneously eliminated loss of bulk polysomes evoked by nutrient starvation and also increased initiation rates of certain mRNAs (*Holmes et al., 2004*; *Coller and Parker, 2005*; *Arribere et al., 2011*). Supporting a direct role in repressing translation, overexpressing Dhh1 or Pat1 in non-starved cells evoked polysome disassembly and reduced the initiation rate of specific mRNAs; and addition of Dhh1 (*Coller and Parker, 2005*) or N-terminally truncated Pat1 (*Nissan et al., 2010*) to yeast extracts inhibited bulk translation and 48 S preinitiation complex (PIC) assembly in vitro. Ribosome profiling studies of *dhh1Δ, pat1Δ* and *pat1Δdhh1Δ* mutants identified hundreds of genes whose transcripts are translationally down-regulated or activated by Dhh1 or Pat1 in nutrient-replete cells (*Jungfleisch et al., 2017*; *Radhakrishnan et al., 2016*; *Zeidan et al., 2018*), which frequently involves cooperation between Dhh1 and Pat1 (*Vijjamarri et al., 2023a*).

Recently, we showed that Pat1 and Dhh1 function with the decapping enzyme in rich medium to repress the abundance or translation of numerous mRNAs encoding proteins required specifically in media containing an alternative carbon or nitrogen source (*Vijjamarri et al., 2023a*; *Vijjamarri et al., 2023b*). These include mitochondrial proteins involved in oxidative phosphorylation (Ox. Phos.) and diverse other proteins known to be transcriptionally repressed by carbon or nitrogen catabolite repression in rich medium.

In this study, we used a multi-omics approach to determine whether Edc3 and Scd6 have largely redundant functions in targeting mRNAs for decapping and attendant degradation, and whether they functionally cooperate with Pat1 or Dhh1 in repressing the abundance or translation of specific mRNAs. We identified a large cohort of mRNAs that are up-regulated in an *scd6Δedc3Δ* double mutant but not in either single mutant lacking only Scd6 or Edc3, without a commensurate increase in transcription of the cognate genes. These transcripts display a strikingly similar pattern of up-regulation in the *dhh1Δ* mutant in the manner predicted if Edc3/Scd6 redundantly recruit Dhh1 to Dcp2 for activation of decapping (*He et al., 2022*). We further observed functional redundancy between Scd6 and Edc3 and extensive cooperation with Dhh1/Pat1 in repressing translation of particular mRNAs, with evidence for interdependent repression by all four decapping factors. Importantly, Edc3/Scd6 contribute to post-transcriptional repression of proteins required for catabolism of non-preferred carbon or nitrogen sources on rich medium, acting collectively to enhance glucose repression, maintain low-level mitochondrial electron transport, and reduce levels of tricarboxylic acid (TCA) and glyoxylate cycle intermediates in glucose-replete cells.

## Results

### Evidence that Scd6 and Edc3 functionally cooperate to control the abundance of many individual mRNAs

To determine whether Scd6 and Edc3 function redundantly in post-transcriptional control of gene expression, we constructed a *scd6Δedc3Δ* double mutant isogenic to the *scd6Δ* and *edc3Δ* single mutants we examined previously (*Zeidan et al., 2018*). Only the double mutant exhibits a marked slow-growth (Slg⁻) phenotype on synthetic complete medium (SC), which was largely complemented by introducing either *SCD6* or *EDC3* on a single copy plasmid (*Figure 1—figure supplement 1A*). Analysis of polysome assembly revealed a ~40% reduction in ratio of polysomes to monosomes (P/M) in the *scd6Δedc3Δ* double mutant, whereas the single mutants showed little (*scd6Δ*) or no (*edc3Δ*) reduction in bulk translation by this assay (*Figure 1—figure supplement 1B*). These results suggest that Scd6 and Edc3 act redundantly to carry out one or more functions required for WT levels of bulk translation and cell growth in nutrient-replete cells.

To examine the effects of the *scd6Δ, edc3Δ*, and *scd6Δedc3Δ* mutations on the abundance and translation of individual mRNAs, we conducted RNA-Seq and ribosome profiling (Ribo-Seq) of the mutant and WT strains following growth in liquid rich medium (YPD) at 30°C (processed data compiled in *Source data 1*). Ribo-Seq entails deep-sequencing of ribosome-protected fragments (RPFs, or ribosome footprints), and cycloheximide was added to the lysates to arrest elongating ribosomes on the mRNA following cell breakage. The ratio of RPF sequencing reads summed over the coding sequences (CDS) to the total mRNA reads from RNA-Seq for the corresponding transcript provides a

measure of translational efficiency (TE) for each mRNA (*Ingolia et al., 2009*). The ribosome profiling and RNA-Seq results between two biological replicates for each strain were highly reproducible with Pearson correlation coefficients (r) ranging between 0.95–1.0 for different pairwise comparisons of replicates (*Figure 1—figure supplement 2A–B*). We employed DESeq2 (*Love et al., 2014*) to identify statistically significant differences in relative mRNA abundance, RPF abundance, or TE for all expressed mRNAs between WT and mutant strains (see Methods for details).

Analysis of the RNA-Seq results identified 81 mRNAs that were significantly up-regulated in the *edc3Δ* mutant vs. WT by >1.5 fold at a false discovery rate (FDR) of <0.05 (dubbed mRNA_up_e3 transcripts; *Figure 1—figure supplement 3A*), and 123 mRNAs reduced in abundance by *edc3Δ* by the same criteria (dubbed mRNA_dn_e3, *Figure 1—figure supplement 3B*). Only 14 mRNAs were up-regulated and only 34 down-regulated by the *scd6Δ* single mutation (respectively, mRNA_up_s6 and mRNA_dn_s6, *Figure 1—figure supplement 3A–B*). Importantly, many more mRNAs were dysregulated by the *scd6Δedc3Δ* double mutation: 741 in the mRNA_up_s6,e3 group and 793 in the mRNA_dn_s6,e3 group (*Figure 1—figure supplement 3A–B*), indicating that the two factors have highly redundant functions in controlling mRNA abundance.

Many yeast mutants with Slg[-] phenotypes, including *pat1Δ, dhh1Δ,* and *dcp2Δ* deletion mutants, exhibit altered expression of most mRNAs belonging to the Environmental Stress Response (ESR) (*O'Duibhir et al., 2014*), which includes ~300 induced (iESR) and ~600 down-regulated (rESR) mRNAs dysregulated in WT cells by various stresses (*Gasch et al., 2000*). In keeping with its Slg[-] phenotype, the *scd6Δedc3Δ* mutation conferred a marked reduction in median expression of rESR mRNAs, and increased expression of the iESR mRNAs, which exceeded in magnitude the changes observed for the slowest growing yeast deletion mutants analyzed previously (*O'Duibhir et al., 2014*; *Figure 1—figure supplement 3C–D*). The two single mutations, by contrast, conferred much smaller changes in ESR mRNAs (*Figure 1—figure supplement 3C–D*). (In all box plots, when notches do not overlap between adjacent boxes, their two medians differ with 95% confidence; and when notches do not overlap 0 in $\log_2$ plots, the median differs significantly from that of all mRNAs, which is invariably close to 1.0.) Consistent with these results, the transcripts up-regulated in the double mutant are enriched for iESR mRNAs (*Figure 1A*), suggesting that the 187 iESR transcripts up-regulated in this strain are responding indirectly to cell stress. The remaining 75% of mRNAs up-regulated in *scd6Δedc3Δ* cells are not iESR mRNAs however (*Figure 1A*), suggesting that their increased abundance arises from eliminating Edc3/Scd6 functions in mRNA decay. Below, we excluded the ESR mRNAs from analyses of mRNA changes in an effort to focus on the transcripts controlled directly by Scd6 and Edc3.

Considering only mRNAs not governed by the ESR, we identified 591 non-iESR mRNAs significantly up-regulated in any of the three mutants (*Figure 1B*; e.g., Non-iESR mRNA_up_s6,e3 designating the 554 mRNAs up-regulated in the double mutant). Examining the ΔRNA values for the majority fraction of the Non-iESR transcripts up-regulated only in the double mutant reveals little change in median abundance in each single mutant but strong up-regulation in the double mutant (*Figure 1B–C*, sectors (iii)), as expected for redundant repressive functions of Scd6 and Edc3. *MDH2*, encoding an enzyme of the glyoxylate cycle, exemplifies a non-iESR transcript up-regulated in both mRNA and RPF abundance exclusively in the double mutant (*Figure 1D*). The small fraction of 27 mRNAs significantly up-regulated in both the *edc3Δ* single mutant and double mutant exhibits only slightly elevated median abundance in the *scd6Δ* single mutant, and only slightly greater up-regulation in the double mutant vs. *edc3Δ* single mutant (*Figure 1B–C*, sectors (ii)), indicating a minimal repressive contribution by Scd6. Interestingly, the small set of 37 mRNAs significantly up-regulated only in the *edc3Δ* single mutant shows reduced rather than increased abundance in the *scd6Δ* single mutant, and lower up-regulation in the double mutant vs. the *edc3Δ* single mutant (*Figure 1B–C*, sectors (i)), suggesting that Scd6 enhances rather than represses these mRNAs, especially in *edc3Δ* cells.

## Evidence that Dhh1 and Pat1 functionally cooperate with Scd6/Edc3 in repressing mRNA abundance

We recently identified a group of 1018 non-iESR mRNAs up-regulated by either *pat1Δ, dhh1Δ,* or *pat1Δdhh1Δ* mutations (*Vijjamarri et al., 2023a*). Importantly, this group is highly enriched for the 591 mRNAs up-regulated by *scd6Δ, edc3Δ,* or *scd6Δedc3Δ* mutations (*Figure 2A*). Indeed, ~70% of the transcripts up-regulated by *scd6Δ/edc3Δ* are also up-regulated in one of the three *pat1Δ/dhh1Δ* mutants (*Figure 2A*, sector (ii)), showing comparable up-regulation in the *scd6Δedc3Δ, pat1Δ,* and

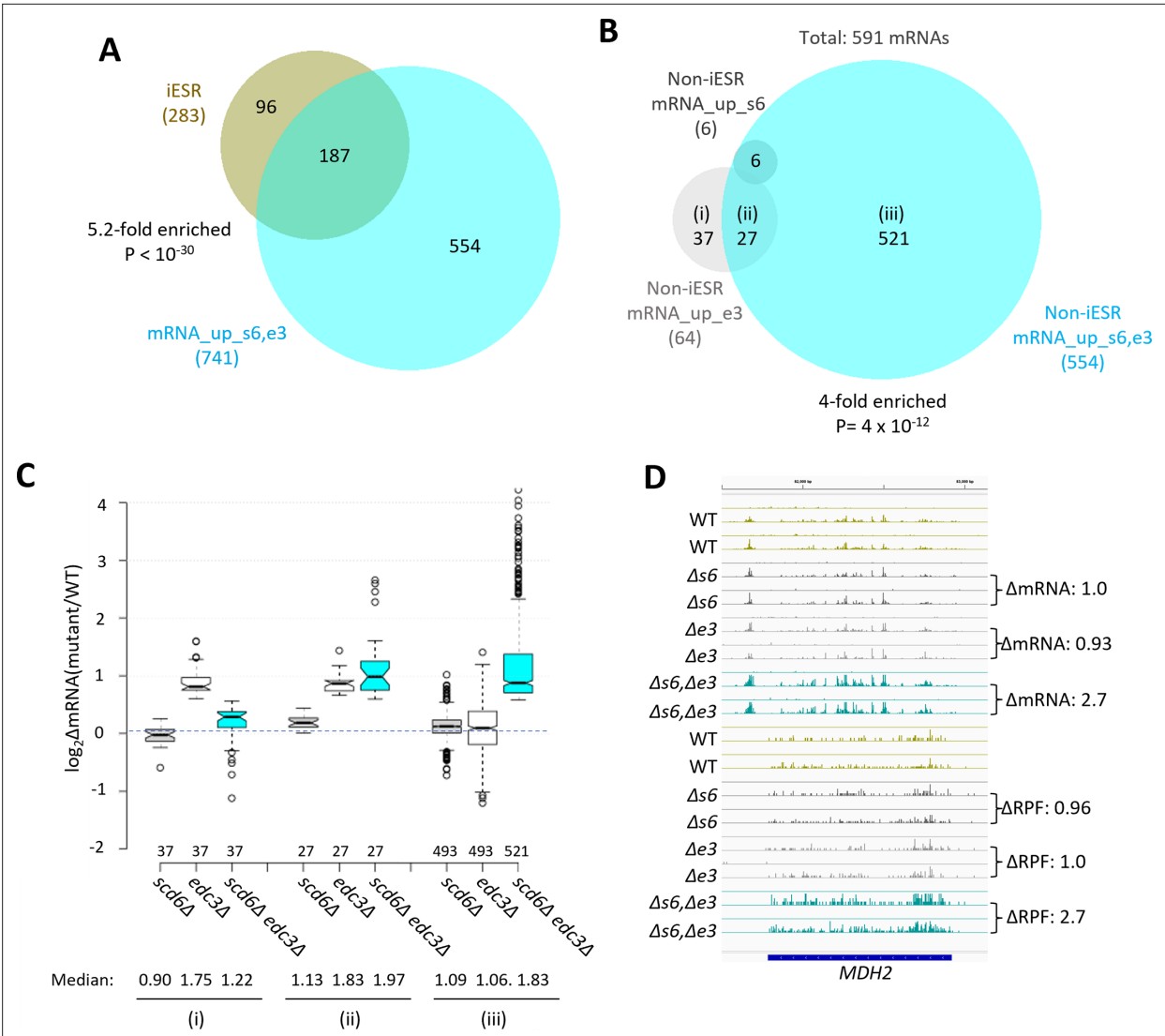

**Figure 1.** Most mRNAs up-regulated in the *scd6Δedc3Δ* mutant are not iESR transcripts and exhibit Scd6/Edc3 functional redundancy in repression of transcript abundance. (**A**) Venn diagram of overlap between the 741 mRNAs up-regulated in the *scd6Δedc3Δ* mutant vs. wild-type (WT) (mRNA_up_ s6,e3) and the 283 induced ESR (iESR) mRNAs, indicating fold-enrichment and p value of overlap determined by the hypergeometric distribution. (**B**) Venn diagram of overlaps involving all 591 non-iESR mRNAs up-regulated in abundance by *scd6Δ* (6 Non-iESR mRNA_up_s6 transcripts), *edc3Δ* (64 mRNA_up_e3 transcripts), or *scd6Δedc3Δ* mutations (554 Non-iESR mRNA_up_s6,e3 transcripts). (**C**) Notched box-plot analyses of $\log_2$ changes in mRNA abundance ($\log_2\Delta$mRNA) determined by DESeq2 analysis between the indicated mutants vs. WT for mRNAs belonging to the specified sectors of the Venn diagram in (**B**). The numbers of mRNAs in each group for which data were obtained are indicated immediately above the x-axis; unlogged median values are indicated for each column at the bottom. (**D**) Gene browser image for *MDH2* showing the mRNA (top 16 tracks) and ribosome-protected fragment (RPF) (bottom 16 tracks) reads measured by parallel RNA-Seq and ribosome profiling analyses for two biological replicates of WT and the indicated mutants with fold-changes in mRNA or RPFs between mutant and WT indicated to the right of each track. In panels **A-D**, results for the *scd6Δ*, *edc3Δ*, and *scd6Δedc3Δ* mutations, abbreviated as *Δs6, Δe3*, and *Δs6,Δe3*, are shown in gray, white or light gray, and cyan, respectively.

The online version of this article includes the following figure supplement(s) for figure 1:

**Figure supplement 1.** Combining *scd6Δ* and *edc3Δ* mutations confers synthetic reductions in cell growth and polysome assembly.

**Figure supplement 2.** Reproducibility among biological replicates of RNA-Seq, Ribo-Seq, External RNA Controls Consortium (ERCC)-normalized RNA-Seq, Rpb1 ChIP-seq, and TMT mass spectrometry (TMT-MS) data.

**Figure supplement 3.** Functional redundancy between Scd6 and Edc3 in controlling mRNA abundance and mobilizing the environmental stress response (ESR).

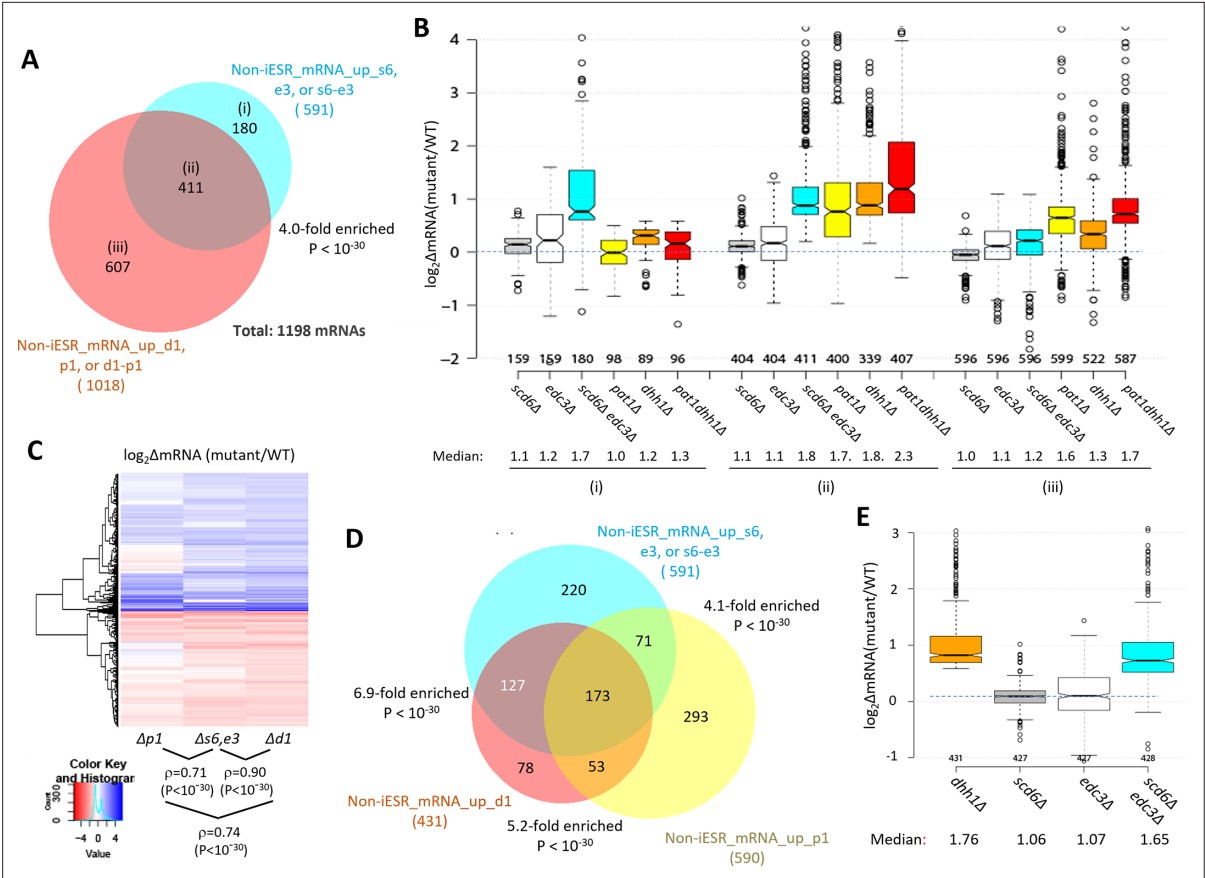

**Figure 2.** Most mRNAs up-regulated in the *scd6Δedc3Δ* mutant are also up-regulated by *dhh1Δ* and *pat1Δ*. (**A**) Venn diagram of overlap between all 591 non-iESR mRNAs up-regulated in abundance by the *scd6Δ*, *edc3Δ*, or *scd6Δedc3Δ* mutations (from **Figure 1B**) and 1018 non-iESR mRNAs up-regulated by the *dhh1Δ*, *pat1Δ*, or *pat1Δdhh1Δ* mutations identified previously (**Vijjamarri et al., 2023a**). Fold enrichments and p value from the hypergeometric distribution are indicated for the overlap. (**B**) Notched box plots of log₂ΔmRNA between the indicated mutants vs. wild-type (WT) for mRNAs in the three sectors specified in (**A**). (**C**) Hierarchical clustering analysis of log₂ΔmRNA values conferred by the indicated mutations vs. WT for 784 of the 794 mRNAs up- or down-regulated in *dhh1Δ* vs. WT cells for which RNA-Seq data was obtained in all four strains and with log₂ΔmRNA values >-5 and <5, conducted with R heatmap.2 function from R 'gplots' library, using default hclust hierarchical clustering algorithm. Spearman coefficients ($\rho$) and associated p values are given for the indicated correlation analyses. (**D**) Venn diagram of overlaps between the 591 non-iESR mRNAs up-regulated by the *scd6Δ/edc3Δ* mutations vs. WT (from **Figure 1B**) and the indicated 431 and 590 non-iESR mRNAs up-regulated by *dhh1Δ* or *pat1Δ* vs. WT, respectively, identified previously (**Vijjamarri et al., 2023a**). The 220, 78, and 293 mRNAs found exclusively in only one of the three sets are indicated in the cyan, red-orange, and yellow sectors, respectively while the 127, 71, and 53 mRNAs shared by two of the three sets and the 173 mRNAs shared by all three sets are indicated in the corresponding regions of overlap. Fold enrichments and p values from the hypergeometric distribution are indicated for the overlaps. (**E**) Notched box plots of log₂ΔmRNA between the indicated mutants vs. WT for the 431 non-iESR mRNAs up-regulated by *dhh1Δ* vs. WT shown in panel (**D**). In panels **A**, **B**, **D**, and **E**, results for mRNAs up-regulated by the *scd6Δ*, *edc3Δ*, and *scd6Δedc3Δ* mutations are shown in gray, white, and cyan, respectively, while those for mRNAs up-regulated by the *pat1Δ*, *dhh1Δ*, and *pat1Δdhh1Δ* mutations are shown in yellow, orange, and red, respectively.

*dhh1Δ* mutants, but little change in the *scd6Δ* and *edc3Δ* single mutants (**Figure 2B(ii)**). These mRNAs generally exhibit the greatest up-regulation in the *pat1Δdhh1Δ* double mutant, indicating cumulative contributions of Dhh1 and Pat1 to their repression, in contrast to the largely redundant roles played by Scd6 and Edc3 in their repression. As expected, the smaller group of 180 mRNAs significantly up-regulated exclusively by *scd6Δ/edc3Δ* (**Figure 2A**, sector (i)) shows strong up-regulation in the *scd6Δedc3Δ* double mutant and they are also appreciably up-regulated by *dhh1Δ* while being largely unaffected by *pat1Δ* (**Figure 2B**, sector (i)). Finally, the majority group of 607 mRNAs significantly up-regulated by only *pat1Δ* or *dhh1Δ* displays the largest increases in the two mutants lacking Pat1, but smaller and similar increases in response to the *scd6Δ/edc3Δ* and *dhh1Δ* mutations (**Figure 2A–B**, sectors (iii)). Overall, these findings suggest that Dhh1 and Pat1 both contribute to repressing the

majority of mRNAs repressed redundantly by Scd6 and Edc3, with Dhh1 contributing more extensively than Pat1.

Further evidence for this last point comes from a k-means clustering analysis of mRNA changes in different mutants for the group of 431 non-iESR mRNAs up-regulated by *dhh1Δ*, which revealed a stronger correlation between the changes conferred by *scd6Δ/edc3Δ* and *dhh1Δ* ($\rho$ =0.90) than between *pat1Δ* with either *dhh1Δ* ($\rho$ =0.74) or *scd6Δ/edc3Δ* ($\rho$ =0.71) (*Figure 2C*). Consistent with this, the *scd6Δ/edc3Δ* mutations up-regulate a considerably larger proportion of the mRNAs significantly up-regulated by *dhh1Δ* (~70%) vs. those up-regulated by *pat1Δ* (~41%) (*Figure 2D*). Furthermore, the majority of mRNAs up-regulated by *dhh1Δ* exhibit redundant repression by Scd6 and Edc3, showing marked up-regulation comparable to that given by *dhh1Δ* itself only in the *scd6Δ/edc3Δ* double mutant (*Figure 2E*). These findings are consistent with the involvement of Edc3/Scd6 in the degradation of most mRNAs targeted by Dhh1, which are more variably and less extensively repressed by Pat1.

## Evidence that Scd6/Edc3 repress mRNA abundance by enhancing decapping/degradation and facilitating Dhh1 association with Dcp2

We next examined whether the mRNAs significantly down-regulated by Edc3/Scd6 are also regulated by the decapping enzyme Dcp1/Dcp2. Supporting this, cluster analysis of mRNA changes revealed that the majority of non-iESR mRNAs up-regulated in the *scd6Δedc3Δ* double mutant are also up-regulated by *dcp2Δ* (*Figure 3A*, blue colors in cols. 1 & 3), with a strong correlation between the abundance changes conferred by these mutations relative to WT ($\rho$ =0.74). In addition, *dcp2Δ* increased the median abundance of this group of mRNAs similarly to that given by *scd6Δedc3Δ* (*Figure 3B*, cols. 1 & 4). These findings implicate decapping by Dcp1:Dcp2 as an important driver of the repression of mRNA levels directed by Scd6/Edc3. The cluster analysis in *Figure 3A* once again reveals greater similarity between the mRNA changes conferred by *scd6Δedc3Δ* and those given by *dhh1Δ* ($\rho$ =0.90) versus *pat1Δ* ($\rho$ =0.69); and *pat1Δ* also confers a smaller median reduction than does *dhh1Δ* for the Scd6/Edc3-down-regulated mRNAs (*Figure 3B*).

Additional support that Scd6/Edc3 target mRNAs for decapping came from evidence that mRNAs up-regulated in the *scd6Δedc3Δ* mutant tend to accumulate in WT cells as decapped isoforms. Following decapping by Dcp1/Dcp2 mRNAs frequently undergo 5' to 3' decay co-translationally, with Xrn1 following behind the last translating ribosome loaded prior to decapping, and such decapped intermediates account for ~12% of all mRNAs in WT cells (*Pelechano et al., 2015*). We reasoned that mRNAs preferentially targeted by Scd6/Edc3 for decapping and attendant degradation by Xrn1 should exhibit a greater than average proportion of decapped intermediates in WT cells. To test this, we conducted cap analysis of gene expression (CAGE) to quantify the abundances of all capped mRNA 5' ends and compare them to total mRNA abundances determined by RNA-Seq conducted in parallel on biological replicates prepared from the WT and *scd6Δedc3Δ* strains described above. Transcript numbers per million reads (TPMs) from CAGE (C) and RNA-Seq (T) were determined and C/T ratios calculated as a proxy for the proportion of capped molecules for each transcript (*Vijjamarri et al., 2023a*; *Source data 1*). (Because the CAGE and RNA-Seq data were normalized separately, the C/T ratios are relative, not absolute, proportions of capped transcripts.) Importantly, the C/T ratios are lower in WT cells for the 498 non-iESR mRNAs up-regulated in the *scd6Δedc3Δ* double mutant compared to all expressed mRNAs (*Figure 3C*, cols. 1 & 5) in the manner expected for mRNAs preferentially targeted for decapping in WT cells. By contrast, the C/T ratios are higher than average for the non-rESR mRNAs that are down-regulated in relative abundance in *scd6Δedc3Δ* cells, as expected for an unusually low degree of decapping in WT (*Figure 3C*, cols 3 & 5). Importantly, these C/T ratios were elevated in *scd6Δedc3Δ* cells to nearly the same level for all three groups of mRNAs, which exceeds the ratios found in WT cells, in the manner expected from eliminating Scd6/Edc3-stimulated decapping (*Figure 3C*, cf. cols. 2, 4, 6). Similar results were obtained for these same groups of dysregulated mRNAs by analyzing our previous CAGE data obtained for the *dhh1Δ* mutant (*Vijjamarri et al., 2023a*; *Figure 1—figure supplement 3E*). These findings are consistent with the notion that impaired decapping dependent on Scd6/Edc3 and Dhh1 is an important driver of increased mRNA abundance in the *scd6Δedc3Δ* mutant.

Independent evidence for this last conclusion was provided by analyzing the codon protection indices (CPI) of mRNAs up-regulated by *scd6Δedc3Δ*, an indicator of co-translational decay by Xrn1.

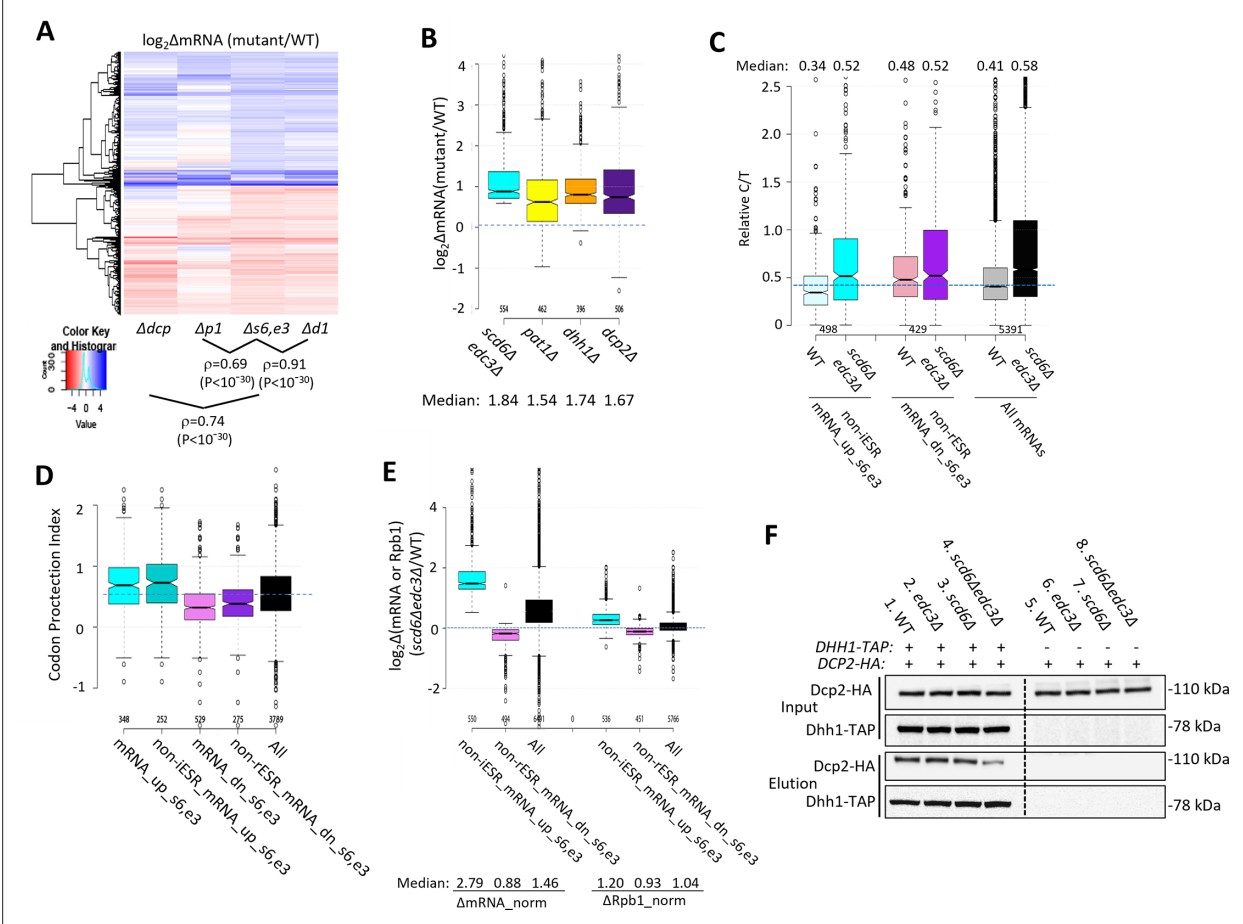

**Figure 3.** Evidence that impaired decapping not increased transcription, drives changes in mRNA abundance in *scd6Δedc3Δ* cells and that Edc3/Scd6 redundantly promote Dhh1 association with Dcp2. (**A**) Hierarchical clustering analysis of $log_2$ΔmRNA values conferred by the indicated mutations vs. wild-type (WT) for 741 of the 1052 mRNAs up- or down-regulated in *scd6Δedc3Δ* vs. WT cells for which RNA-Seq data was obtained in all five strains and with $log_2$ΔmRNA values >-5 and <5, conducted as in ***Figure 2C***, showing Spearman coefficients and p values for indicated correlations. (**B**) Notched box-plots of $log_2$ΔmRNA between the indicated mutants vs. WT for the 554 non-iESR mRNAs up-regulated by *scd6Δedc3Δ* vs. WT (shown in ***Figure 1B***). (**C**) Ratios of capped to total mRNA abundance in transcript numbers per million reads (TPMs) (Relative C/T) in WT or *scd6Δedc3Δ* cells plotted for all 5391 mRNAs, the 498 non-iESR mRNA_up_s6,e3, or 429 non-rESR mRNA_down_s6,e3 transcripts dysregulated by *scd6Δedc3Δ* vs. WT. (**D**) Notched box plots of the codon-protection index (CPI) for all mRNAs or for the sets of mRNAs up- or down-regulated by *scd6Δedc3Δ* vs. WT, including or excluding ESR transcripts, as indicated. (**E**) Notched box-plots showing $log_2$ changes in absolute mRNA abundance from External RNA Controls Consortium (ERCC) spike-in normalized RNA-Seq (left) or absolute Rpb1 occupancies averaged over the coding sequences (CDSs) from *S. pombe* chromatin spike-in normalized Rpb1 ChIP-Seq (right) in *scd6Δedc3Δ* vs. WT cells for all mRNAs or the 554 or 526 non-ESR mRNAs up- or down-regulated, respectively, by *scd6Δedc3Δ* vs. WT. (**F**) Co-immunoprecipitation analysis of Dhh1-Dcp2 association in yeast cell extracts. Transformants of *DHH1-TAP* strains H5695 (WT), H5696 (*edc3Δ*), H5697 (*scd6Δ*), and H5698 (*edc3Δ scd6Δ*) and the parental untagged strains all harboring single-copy plasmid pAK133 expressing HA-tagged Dcp2 were cultured in SC-Ura at 30°C and whole cell extracts were incubated with IgG sepharose beads to purify Dhh1-TAP and associated proteins. Washed beads were eluted by boiling in SDS loading buffer and eluates and input extracts were resolved in parallel by SDS-PAGE and subjected to immunoblot analysis to detect the tagged proteins. Immune complexes were visualized with enhanced chemiluminescence.

The online version of this article includes the following source data and figure supplement(s) for figure 3:

**Source data 1.** Relative and spike-in normalized Rpb1 occupancies from ChIP-seq analysis.

**Source data 2.** External RNA Controls Consortium (ERCC) spike-in normalized RNA-Seq data.

**Source data 3.** Figures of the uncropped blots with the relevant bands labelled used to prepare *Figure 3F*.

**Source data 4.** Original images of the full uncropped, unedited blots used to prepare *Figure 3F*.

**Figure supplement 1.** Supporting information for measurements of transcription and abundance of transcripts dysregulated in the *scd6Δedc3Δ* double mutant.

**Figure supplement 2.** Biological replicates of co-immunoprecipitation analysis of Dhh1-Dcp2 association in yeast cell extracts.

**Figure supplement 2—source data 1.** Figures of the uncropped blots with the relevant bands labelled used to prepare *Figure 3—figure supplement*

*Figure 3 continued*

2.

**Figure supplement 2—source data 2.** Original images of the full uncropped, unedited blots used to prepare *Figure 3—figure supplement 2*.

**Figure supplement 3.** Average median codon optimality scores and average median translational efficiency (TE) values in wild-type (WT) cells for mRNAs repressed in abundance by Scd6/Edc3 or Dhh1/Pat1.

Decapped degradation intermediates exhibit three-nucleotide periodicity generated by precise Xrn1 cleavage up to the last translating ribosome at the 5' end of mRNA, and the CPI quantifies the prevalence of such intermediates for each mRNA (*Pelechano et al., 2015*). Importantly, the mRNA_up_s6,e3 transcripts (whether including or excluding ESR mRNAs) exhibit higher than average median CPIs, indicating a greater than average involvement of decapping and co-translational degradation by Xrn1 in their decay, whereas mRNA_dn_s6,e3 transcripts exhibit lower than average CPI values, consistent with an alternative degradation pathway controlling their abundance *scd6Δedc3Δ* (*Figure 3D*).

To determine whether increased transcription contributes to increased abundance of mRNAs up-regulated by *scd6Δedc3Δ*, we performed ChIP-Seq analysis of Rpb1 to measure RNA Polymerase II (Pol II) occupancies averaged across the CDS of every gene, obtaining highly reproducible results across replicates (*Figure 1—figure supplement 2D* and *Figure 3—figure supplement 1A*). To quantify absolute changes in Pol II occupancies, *S. pombe* chromatin was added to each *S. cerevisiae* chromatin sample prior to immunoprecipitation. To measure absolute changes in mRNA abundance, we re-analyzed the RNA-Seq data taking into account the recovery of External RNA Controls Consortium (ERCC) transcripts that were added to each total RNA sample prior to preparation of cDNA libraries, yielding highly reproducible normalized values among replicates (*Figure 1—figure supplement 2C*). Interrogating the non-ESR mRNAs up- or down-regulated by *scd6Δedc3* revealed a 2.79-fold increase in median normalized mRNA abundance for the non-iESR mRNA_up_s6,e3 group that was associated with only a 1.2-fold increase in normalized Rpb1 occupancies (*Figure 3E*, cols. 1 & 4). The down-regulated non-rESR mRNA_dn_s6,e3 transcripts showed 12% and 7% decreases in spike-in normalized median mRNA abundance and Rpb1 occupancies, respectively (*Figure 3E*, cols. 2 & 5). Because the majority of total RNA is rRNA, the ERCC normalization effectively yields the abundance of each mRNA relative to rRNA, and the ribosome content per cell is expected to be reduced in the *scd6Δedc3Δ* mutant owing to repression of rESR mRNAs encoding ribosomal proteins and biogenesis factors (*Figure 1—figure supplement 3C*). We showed previously that the isogenic *dcp2Δ* mutant displays an ESR response of similar magnitude and a 30% reduction in ribosomal subunits per cell compared to the same WT examined here (*Vijjamarri et al., 2023b*). Assuming a similar reduction in ribosome abundance in *scd6Δedc3Δ* cells, the absolute changes in mRNA per cell conferred by *scd6Δedc3Δ* are expected to be 0.7-fold of the ERCC-normalized values given in *Figure 3E*, yielding fold changes of 2.0 and 0.62 for the mRNA_up_s6,e3 and mRNA_dn_s6,e3 groups, respectively. Because these predicted changes in mRNA per cell still differ substantially from the corresponding changes in normalized Rpb1 occupancies of 1.2 and 0.93, respectively, there is only a small contribution of altered transcription to the altered abundance of transcripts dysregulated in the double mutant.

Our findings above that the *scd6Δedc3Δ* and *dhh1Δ* mutations up-regulate highly similar sets of mRNAs (*Figure 2C*) is consistent with a previous proposal that recruitment of Dhh1 to Dcp2 can be mediated interchangeably by Edc3 or Scd6 bound to the same site in the Dcp2 CTT (*He et al., 2022*). To test this model, we asked whether the association of Dhh1 with Dcp2 in cell extracts is impaired only when both Edc3 and Scd6 are eliminated. Indeed, co-immunoprecipitation of HA-tagged Dcp2 with TAP-tagged Dhh1 was reduced in *scd6Δedc3Δ* cells but not in either single mutant in replicate experiments (*Figure 3F*, lanes 2–4; *Figure 3—figure supplement 2(i-ii)*). The residual association in the double mutant suggests that Dhh1 can interact with Dcp2 either directly or via Pat1 in addition to the interaction mediated by Scd6/Edc3, which might explain why a small fraction of mRNAs are up-regulated by *dhh1Δ* but not by *scd6Δedc3Δ* (*Figure 2C*).

## Slow rates of translation initiation or elongation generally do not dictate preferential decapping/decay by the decapping activators

Codon non-optimality has been linked with Dhh1-mediated mRNA decay partly by demonstrating that the sTAI values of mRNAs, which quantify their overall codon optimality (*Sabi and Tuller, 2014*), are inversely correlated with the changes in mRNA abundance observed in *dhh1Δ* versus WT cells (*Radhakrishnan et al., 2016*). Similarly, analyzing the mRNA changes conferred by the *scd6Δedc3Δ* mutation for all non-ESR transcripts reveals a small but statistically significant negative correlation with sTAI values (Pearson r of –0.085, $p = 2 \times 10^{-9}$) similar to that observed for the *dhh1Δ* mutation ($r = -0.071$, $p = 6 \times 10^{-6}$) (*Figure 3—figure supplement 3A*, cyan vs. orange), indicating a tendency for mRNAs with lower codon optimality/sTAI scores to show greater increases in relative abundance in response to *scd6Δedc3Δ* or *dhh1Δ*. However, the 591 non-iESR mRNAs up-regulated by the *scd6Δ/edc3Δ* mutations (from *Figure 1B*) have a median sTAI value (0.35) nearly identical to that of all nonESR mRNAs, which is also the case for the group of 1018 mRNAs up-regulated by the *pat1Δ* or *dhh1* mutations (*Vijjamarri et al., 2023a*; *Figure 3—figure supplement 3B*). Similar results were obtained for other metrics of codon optimality, tAI and average CSC (*Figure 3—figure supplement 3B*), suggesting that poor codon optimality is not a key property defining the mRNAs repressed most extensively by these decapping activators. It has also been proposed that competition between translation initiation and mRNA decay, rather than codon optimality and elongation, is a major determinant of mRNA stability in yeast (*Muhlrad et al., 1995*; *LaGrandeur and Parker, 1999*; *Schwartz and Parker, 1999*; *Chan et al., 2018*). However, the mRNAs up-regulated in the *scd6Δ/edc3Δ* or *pat1Δ/dhh1* mutants have slightly greater than average median translational efficiencies (TEs) in WT cells (*Figure 3—figure supplement 3C*), as determined by ribosome profiling experiments described below. Thus, neither slow rates of translation initiation nor pausing during elongation at non-optimal codons appears to dictate preferential targeting of the mRNAs most strongly repressed by these four decapping activators.

## Scd6 and Edc3 act redundantly and cooperate more extensively with Dhh1 than Pat1 in controlling translation of individual mRNAs

The changes in mRNA abundance described above were highly correlated with changes in RPF abundance determined by Ribo-Seq analysis for the *scd6Δedc3Δ* mutant compared to WT for all expressed transcripts, with an r value of 0.81 (*Figure 1—figure supplement 2F*), providing strong mutual validation of the RNA-Seq and Ribo-Seq data for these strains. Nevertheless, the fact that this correlation is weaker than that observed between biological replicates of RNA-Seq or Ribo-Seq data (r values >0.95, *Figure 1—figure supplement 2A–B*) suggests that certain mRNAs exhibit altered translational efficiencies in the double mutant. Indeed, DESeq2 analysis of the Ribo-Seq data identified 184 mRNAs showing TE increases of >1.41 fold at FDR <0.10 in the *scd6Δedc3Δ* strain (dubbed TE_up_s6,e3) but only 42, or a single mRNA, in the *scd6Δ* and *edc3Δ* single mutants, respectively (*Figure 4A*), suggesting that Scd6 and Edc3 have overlapping functions in repressing the translation of particular mRNAs. Supporting this, a group of 169 mRNAs translationally up-regulated exclusively in the *scd6Δedc3Δ* double mutant exhibits only modest TE increases in the two single mutants (*Figure 4A–B*, sectors (iii)). *SPI1*, encoding a cell wall protein, is a representative transcript displaying increased mRNA abundance coupled with an even greater increase in RPF abundance that confers a TE increase of ~ threefold only in the double mutant (*Figure 4C*). As expected, the 28 mRNAs showing substantial TE increases only in *scd6Δ* cells show no increase in median TE in the *edc3Δ* strain (*Figure 4A–B*, sectors (i)) and thus appear to be translationally repressed by Scd6 alone.

We explored next whether the increased ribosome occupancies observed in the *scd6Δedc3Δ* mutant are associated with increased protein synthesis by conducting TMT mass spectrometry (TMT-MS) of total cell proteins to obtain ratios of peptide abundance in the *scd6Δedc3Δ* strain vs. WT for >4000 different proteins (*Figure 1—figure supplement 2E*). Importantly, significant correlations exist between changes in protein abundance and RPFs across the translatome (*Figure 4D*). Moreover, mRNAs showing increased or decreased RPFs in *scd6Δedc3Δ* vs. WT cells (>1.5 fold, FDR <0.05, Ribo_up and Ribo_dn) likewise exhibit increased or decreased median protein abundances determined by TMT-MS (*Figure 4E*). These results suggest that increased ribosome occupancies measured by Ribo-Seq, which could occur by increases in mRNA, TE, or both, are generally associated with increased synthesis of the encoded proteins in *scd6Δedc3Δ* cells.

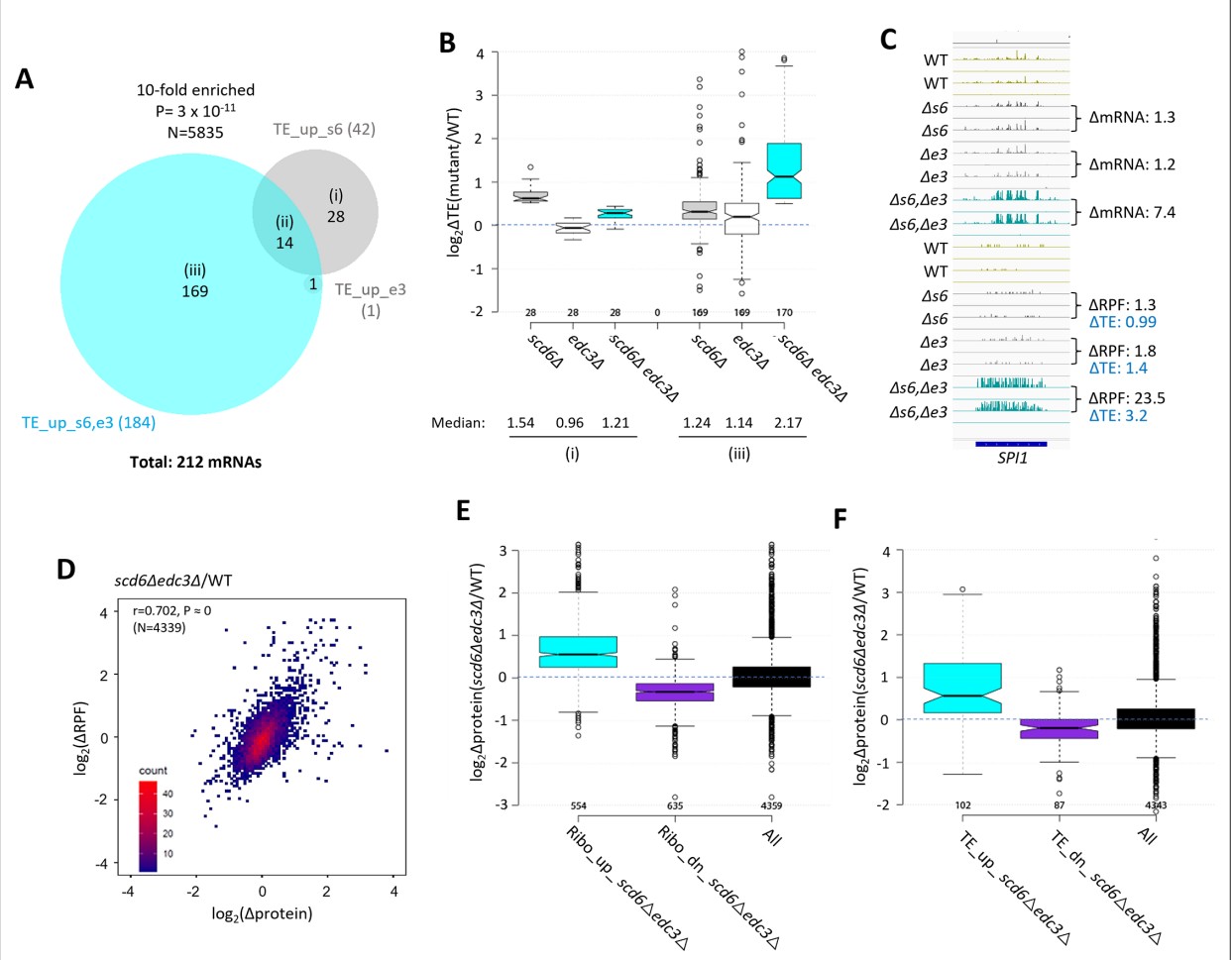

**Figure 4.** Most mRNAs translationally up-regulated in *scd6Δedc3Δ* cells exhibit Scd6/Edc3 functional redundancy for repressing translational efficiency (TE) and show correlated changes in TE and protein abundance. (**A**) Venn diagram of overlap between the 184 and 42 mRNAs in the TE_up groups identified in the *scd6Δedc3Δ* vs. wild-type (WT) or *scd6Δ* vs. WT comparisons, respectively. Fold-enrichment and p value from the hypergeometric distribution are indicated for the overlap. (**B**) Notched box-plots of log$_2$ΔTE values between the indicated mutants vs. WT for the mRNAs belonging to sectors (i) or (iii) of the diagram in (**A**). (**C**) Gene browser image for *SPI1* presented as in *Figure 1D*, except also giving the TE changes for each mutant vs. WT on the lower right. (**D**) Density scatterplot of log$_2$ΔRPF values measured by ribosome profiling vs. log$_2$Δprotein values measured by TMT mass spectrometry (TMT-MS) for 4339 mRNAs for which data were obtained in both analyses, indicating the Pearson correlation coefficient (r) and p-value of the correlation. (**E–F**) Notched box-plots of log$_2$Δprotein values from TMT-MS analysis between the *scd6Δedc3Δ* mutant vs. WT for the 843 and 839 mRNAs belonging to the Ribo_up or Ribo_down groups, respectively (**D**), or the 184 and 152 TE_up or TE_down mRNA groups (**E**) determined for the *scd6Δedc3Δ* mutant vs. WT, or for all mRNAs, for which TMT-MS data was obtained.

It was possible that translational repression by Edc3/Scd6 generally occurs by slowing elongation, leading to increased ribosome densities (RPF/mRNA ratios, i.e. calculated TEs) in WT cells and decreased TE values in *scd6Δedc3Δ* cells that would be associated with increased protein expression. In this scenario, changes in TEs would be inversely associated with changes in protein expression. Instead, mRNAs showing increased or decreased TEs in the double mutant (TE_up_s6,e3 and TE_dn_s6,e3 transcripts defined above) also exhibit increased or decreased protein expression in *scd6Δedc3Δ* vs. WT cells (*Figure 4F*), implying that Scd6/Edc3 generally influence translation at the initiation step.

We previously identified 274 mRNAs whose TEs are up-regulated in the *pat1Δ*, *dhh1Δ*, or *pat1Δdhh1Δ* mutants by the same criteria employed here (>1.41 fold at FDR <0.10) (*Vijjamarri et al., 2023a*). Interestingly, these mRNAs overlap significantly with those translationally up-regulated in the single or double *scd6Δ/edc3Δ* mutants (*Figure 5A*). The 76 mRNAs common to both groups show larger TE increases in the *scd6Δedc3Δ* mutant compared to the 136 transcripts up-regulated

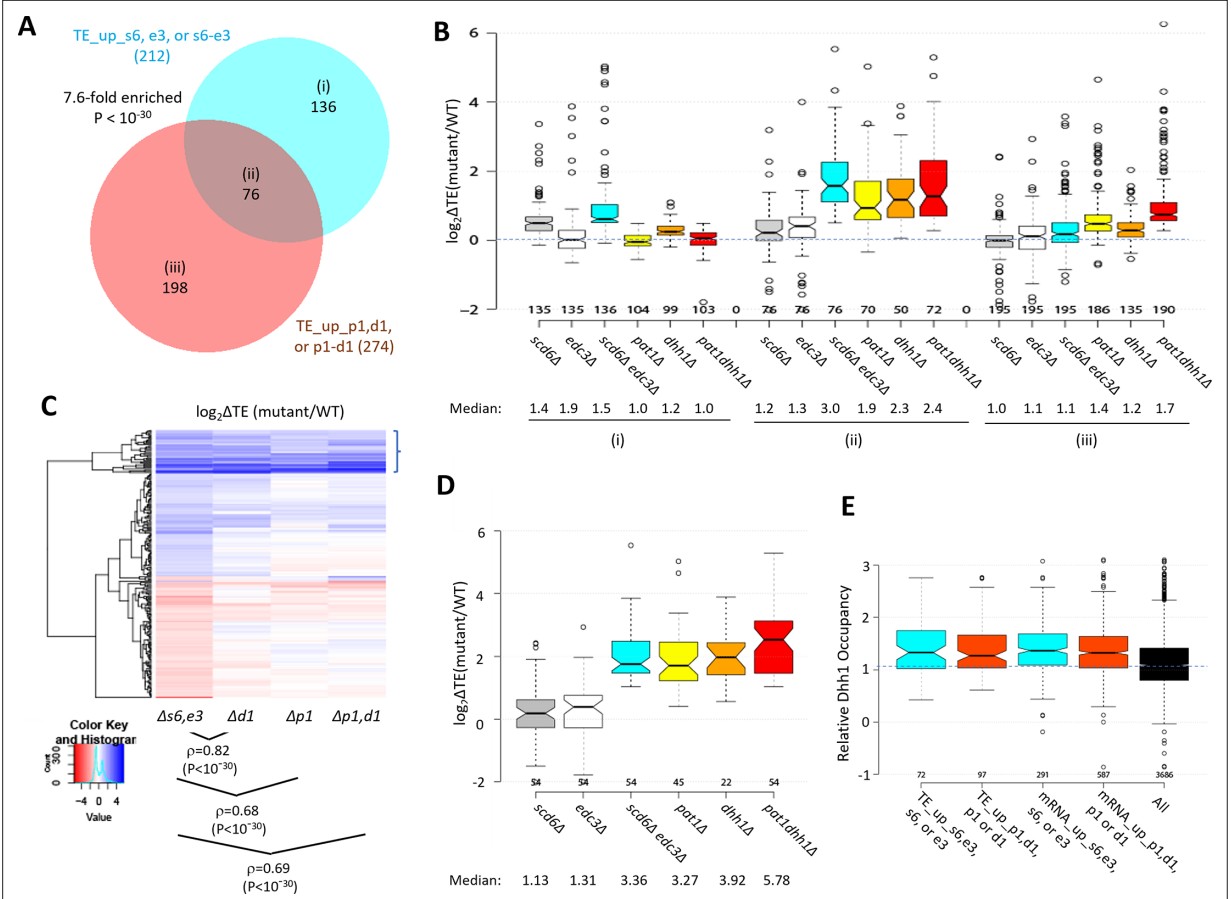

**Figure 5.** Most mRNAs translationally up-regulated in the *scd6Δedc3Δ* mutant are also translationally up-regulated by *dhh1Δ* or *pat1Δ*. (**A**) Venn diagram of overlap between all 212 mRNAs translationally up-regulated by *scd6Δ*, *edc3Δ*, or *scd6Δedc3Δ* (defined in **Figure 4**) or all 274 mRNAs translationally up-regulated by *dhh1Δ*, *pat1Δ*, or *pat1Δdhh1Δ* vs. wild-type (WT) identified previously (**Vijjamarri et al., 2023a**). Fold-enrichment and p value from the hypergeometric distribution are indicated for the overlap. (**B**) Notched box plots of log$_2$ΔTE values between the indicated mutants vs. WT for the mRNAs belonging to the specified sectors of the diagram in (**A**). (**C**) Hierarchical clustering analysis of log$_2$ΔTE values conferred by the indicated mutations vs. WT for 222 of the 336 mRNAs translationally up- or down-regulated in *scd6Δedc3Δ* vs. WT cells for which RNA-Seq and Ribo-Seq data were obtained in all five strains and with log$_2$ΔTE values >-5 and <5 conducted as in **Figure 2C**, including the Spearman coefficients ($\rho$) and p values for the indicated correlations. (**D**) Notched box-plots of log$_2$ΔTE values between the indicated mutants vs. WT for the 54 mRNAs showing >2 fold translational efficiency (TE) increases conferred by both *scd6Δedc3Δ* and *pat1Δdhh1Δ* mutations vs. WT. (**E**) Relative Dhh1 occupancies from the Dhh1 RIP-seq experiments of **Miller et al., 2018** for the 212 and 274 mRNAs identified as TE_up in the *scd6Δ*, *edc3Δ*, or *scd6Δedc3Δ* mutants, or the *dhh1Δ*, *pat1Δ*, or *pat1Δdhh1Δ* mutants, vs. WT, respectively (cols 1–2), or for the 591 and 1018 mRNAs identified as mRNA_up in either the *scd6Δ*, *edc3Δ*, or *scd6Δedc3Δ* mutants, or the *dhh1Δ*, *pat1Δ*, or *pat1Δdhh1Δ* mutants, vs. WT, respectively (cols. 3–4).

The online version of this article includes the following figure supplement(s) for figure 5:

**Figure supplement 1.** Properties of mRNAs translationally repressed by Scd6/Edc3 or Dhh1/Pat1.

exclusively in the *scd6Δ/edc3Δ* mutants (**Figure 5A–B**, sectors (ii) vs. sectors (i), cyan data). These 76 mRNAs also show marked TE increases in the *dhh1Δ*, *pat1Δ*, and *pat1Δdhh1Δ* mutants (**Figure 5B(ii)**), indicating that all four decapping activators are required for efficient translational repression of these mRNAs in WT cells. The 136 mRNAs significantly up-regulated only in the *scd6Δ/edc3Δ* mutants show a moderate increase in median TE in *dhh1Δ* cells, but little response to *pat1Δ* (**Figure 5B(i)**). The 198 mRNAs up-regulated only in the *pat1Δ/dhh1Δ* mutants show only a small TE increase in the *scd6Δedc3Δ* strain and cumulative TE increases in the *pat1Δdhh1Δ* double mutant (**Figure 5B(iii)**). In summary, similar to our findings for repression of mRNA abundance, efficient translational repression of certain mRNAs requires the combined functions of Pat1, Dhh1, and either Scd6 or Edc3, whereas other mRNAs are translationally repressed by either Edc3/Scd6 or Pat1 with appreciable contributions from Dhh1.

Greater cooperation of Scd6/Edc3 with Dhh1 vs. Pat1 in controlling translation was again revealed by clustering analysis of TE changes in different mutants for the mRNAs exhibiting altered TEs in the *scd6Δ/edc3Δ* mutants, with greater similarity between TE changes conferred by *scd6Δedc3Δ* vs. *dhh1Δ* ($\rho$ =0.82) compared to *scd6Δedc3Δ* vs. *pat1Δ* ($\rho$ =0.68) or *scd6Δedc3Δ* vs. *pat1Δdhh1Δ* ($\rho$ =0.69) cells (**Figure 5C**). However, as noted above, the group showing the strongest up-regulation of TEs in the *scd6Δedc3Δ* double mutant tends to be highly up-regulated in all three of the other mutants lacking Dhh1 or Pat1 (bracketed mRNAs at the top of **Figure 5C**), as shown in **Figure 5D** for a group of 54 mRNAs whose TEs are up-regulated by 2.0-fold or more in both the *scd6Δedc3Δ* and *pat1Δdhh1Δ* double mutants. Thus, although Scd6/Edc3 appear to cooperate more extensively with Dhh1 than Pat1 in translational control, the subset of 50–60 mRNAs exhibiting the strongest repression by Scd6/Edc3 or Pat1/Dhh1 requires the concerted functions of Pat1, Dhh1, and either Scd6 or Edc3 to achieve their strong translational repression in WT cells.

To assess whether mRNAs translationally repressed by the decapping activators are also targeted for degradation by these factors, we examined mRNA changes for the groups of mRNAs exhibiting TE up-regulation in the *scd6Δ/edc3Δ* or *pat1Δ/dhh1Δ* mutants (sectors (i) to (iii) of **Figure 5A**). The mRNAs showing increased TEs only in the *pat1Δ/dhh1Δ* strains show little change in mRNA abundance in these mutants (**Figure 5—figure supplement 1A(iii)**), indicating selective repression of translation vs degradation. Similarly, mRNAs translationally up-regulated only in the *scd6Δ/edc3Δ* mutants show little change in mRNA abundance in the *scd6Δedc3Δ* strain and only a modest up-regulation in the *pat1Δ/dhh1Δ* mutants (**Figure 5—figure supplement 1A(i)**). In contrast, the transcripts showing increased TEs in all of the mutants also show markedly increased abundance in the same mutants (**Figure 5—figure supplement 1A(ii)**). Thus, coupled repression of translation and abundance by the decapping factors occurs only for the subset of transcripts exhibiting concerted translational repression by all four factors. In contrast, the much larger groups of transcripts defined above showing increased mRNA abundance in the mutants (described in **Figure 2A**) exhibit little change in median TE in all four mutants (**Figure 5—figure supplement 1B**), suggesting that most mRNAs targeted for enhanced turnover do not exhibit translational repression of the undegraded transcripts remaining in WT cells.

We next examined the native translational efficiencies of mRNAs translationally repressed by Scd6/Edc3 by interrogating the Ribo-Seq data for WT cells on rich medium. The mRNAs translationally repressed by Scd6/Edc3 in concert with Pat1/Dhh1 tend to be poorly translated in WT, having a substantially lower median relative TE compared to all mRNAs (0.24 vs 0.95) (**Figure 5—figure supplement 1C**, col. 2), thus resembling the mRNAs translationally repressed exclusively by Dhh1/Pat1 (in col. 3). In contrast, the transcripts translationally repressed exclusively by Edc3/Scd6 are generally well translated, exhibiting a ~2.5 fold greater than average median TE in WT cells (**Figure 5—figure supplement 1C**, col. 1). Consistent with this distinction, gene ontology (GO) analysis revealed that products of the 136 mRNAs translationally repressed exclusively by Scd6/Edc3 are enriched for cytoplasmic or mitochondrial ribosomal proteins ($p$=1 ×10$^{-7}$), whereas products of the 76 transcripts repressed interdependently by Scd6/Edc3/Pat1/Dhh1 are enriched for factors involved in fermentation ($p$=2 ×10$^{-7}$), carbohydrate metabolism ($p$=1 ×10$^{-6}$) or metabolism of non-protein amino acids ($p$=3 ×10$^{-6}$). Thus, Scd6/Edc3 translationally repress distinct groups of mRNAs depending on whether Dhh1/Pat1 participate in the repression.

We showed previously (**Zeidan et al., 2018**) that mRNAs up-regulated by *dhh1Δ* are enriched for Dhh1 protein association in vivo, as judged by RIP-Seq analysis of yeast mRNAs using antibodies against Dhh1 (**Miller et al., 2018**), thus providing evidence for a direct role of Dhh1 in decapping/degradation of these transcripts. The same observation was made for the mRNAs up-regulated by *pat1Δ,* consistent with widespread cooperation between Dhh1 and Pat1 in repressing mRNA abundance (**Vijjamarri et al., 2023a**). Interestingly, here we observed greater than average Dhh1 occupancies for all four mRNA groups defined above (in **Figure 2A** or **Figure 5A**) that are up-regulated in transcript abundance or TE by the *scd6Δ/edc3Δ* or *dhh1Δ/pat1Δ* mutations (**Figure 5E**). This finding is consistent with the fact that Dhh1 contributes to repressing the abundance or TE of the majority of transcripts in these groups (**Figures 2B and 5B**, orange data). It is possible that Dhh1 is recruited to most of these mRNAs in a complex with Dcp1:Dcp2 and other decapping activators (**He et al., 2022**), but contributes differentially to their degradation or translational repression.

# Scd6/Edc3 post-transcriptionally repress mRNAs encoding enzymes for respiration and catabolism of non-glucose carbon sources in rich medium

Recently, we reported that mRNAs up-regulated in abundance or TE in the *dcp2Δ*, *pat1Δ*, and *dhh1Δ* mutants are enriched for mitochondrial proteins that function in Ox. Phos. (*Vijjamarri et al., 2023a*; *Vijjamarri et al., 2023b*). GO analysis led to the same finding here for the non-iESR mRNAs up-regulated in abundance by the *scd6Δ/edc3Δ* mutations, including many of the same categories of mitochondrial functions enriched among the mRNAs up-regulated by *dhh1Δ/pat1Δ* mutations (*Figure 6—figure supplement 1A–B*, green type). GO analysis of genes showing increased ribosome occupancies (RPFs), indicating either increased mRNA abundance or TE, confirms that the *scd6Δ/edc3Δ* mutations derepress the translation of Ox. Phos. gene transcripts (*Figure 6—figure supplement 1C–D*, green). Examining a collection of Ox. Phos. genes functioning in electron transport, the TCA cycle, or mitochondrial ATP synthase reveals that the *scd6Δedc3Δ* and *pat1Δdhh1Δ* double mutations up-regulate expression of these genes primarily at the level of mRNA abundance, with small additional increases in TE (*Figure 6A*). Western blot analysis revealed increased expression of four Ox. Phos. proteins (Qcr8, Atp20, Idh1, and Sdh4), two proteins (Cox14 and Cox20) involved in cytochrome c oxidase assembly, and mitochondrial cytochrome b2 (Cyb2) required for lactate utilization in *scd6Δedc3Δ* cells, relative to Gcd6 examined as loading control (*Figure 6B–C*). Results similar to these were obtained for the same mitochondrial proteins in the *dhh1Δ* and *pat1Δ* mutants (*Vijjamarri et al., 2023a*) and for a subset in *dcp2Δ* cells (*Vijjamarri et al., 2023b*). Importantly, we also observed increased expression of Cox2, a mitochondrially encoded subunit of cytochrome *c* oxidase, terminal enzyme of the mitochondrial ETC, whose expression correlates with mitochondrial activity (*Vengayil et al., 2024*). Except for *dcp2Δ*, all the decapping mutants had significantly increased Cox2 protein (*Figure 6D*), suggesting an increase in ETC activity in these mutants. Up-regulation of mRNA and RPF abundance in *scd6Δedc3Δ* and *pat1Δdhh1Δ* double mutants also occurred for enzymes of the glyoxylate cycle (*Figure 6E*), which catalyze certain reactions of the TCA cycle in the cytoplasm to support gluconeogenesis during respiratory growth on two-carbon compounds; and the increased expression of one such enzyme, Cit2, was confirmed by Western analysis (*Figure 6B–C*). Both Ox. Phos. and the glyoxylate cycle normally operate at low levels in yeast growing with abundant glucose, as in our experiments, suggesting that Scd6/Edc3 cooperate with Dhh1/Pat1 to help suppress these pathways in glucose-replete cells.

Consistent with post-transcriptional control of Ox. Phos. genes, the median relative RPF levels were up-regulated in the *scd6Δedc3Δ* mutant substantially more than the increases in relative Pol II occupancies observed at the cognate genes by ChIP-Seq analysis of Rpb1 (*Figure 6—figure supplement 2B*, cols. 1–2). Moreover, the transcription factors responsible for induction of Ox. Phos. genes, the Hap2/Hap3/Hap4/Hap5 complex, were not activated in *scd6Δedc3Δ* cells: expression of a *CYC1-lacZ* reporter activated by this complex (*Forsburg and Guarente, 1989*) was not elevated in the *scd6Δ/edc3Δ* mutants in glucose-containing medium (*Figure 6F(ii)*) but showed the expected induction in WT cells grown with glycerol/ethanol versus glucose (*Figure 6F(i)*; *Broach, 2012*). These findings support the notion that reduced decapping stabilizes Ox. Phos. gene transcripts in *scd6Δedc3Δ* cells.

In addition to Ox. Phos., GO analysis of genes up-regulated in RPF abundance in the *scd6Δ/edc3Δ* mutants revealed enrichment for utilization of alternative carbon sources (*Figure 6—figure supplement 1C*), as observed recently for *dhh1Δ/pat1Δ* mutants (*Vijjamarri et al., 2023a*). Consistent with this, 83 carbon catabolite down-regulated (CCR) genes, known to be glucose-repressed or activated by transcription factors Adr1 or Cat8 (*Young et al., 2003*; *Tachibana et al., 2005*), exhibit increased translation (RPFs) largely through increased mRNA abundance in both *scd6Δedc3Δ* and *pat1Δdhh1Δ* double mutants (*Figure 6—figure supplement 2A*). These genes encode enzymes for β-oxidation of fatty acids in addition to the glyoxylate cycle, which allows cells to synthesize precursors that feed into gluconeogenesis or amino acid biosynthesis, or produce acetyl-CoA and generate NADH by respiration on non-fermentable carbon sources (*Young et al., 2003*). The CCR genes exhibit larger increases in RPFs compared to Pol II occupancies at the cognate genes in *scd6Δedc3Δ* vs. WT cells (*Figure 6—figure supplement 2B*, cols. 3–4), consistent with post-transcriptional repression by Scd6/Edc3. Supporting this, an *ADH2-lacZ* reporter transcriptionally induced by activated Adr1 (*Sloan et al., 1999*) displayed the expected large induction in our WT strain cultured with glycerol/ethanol

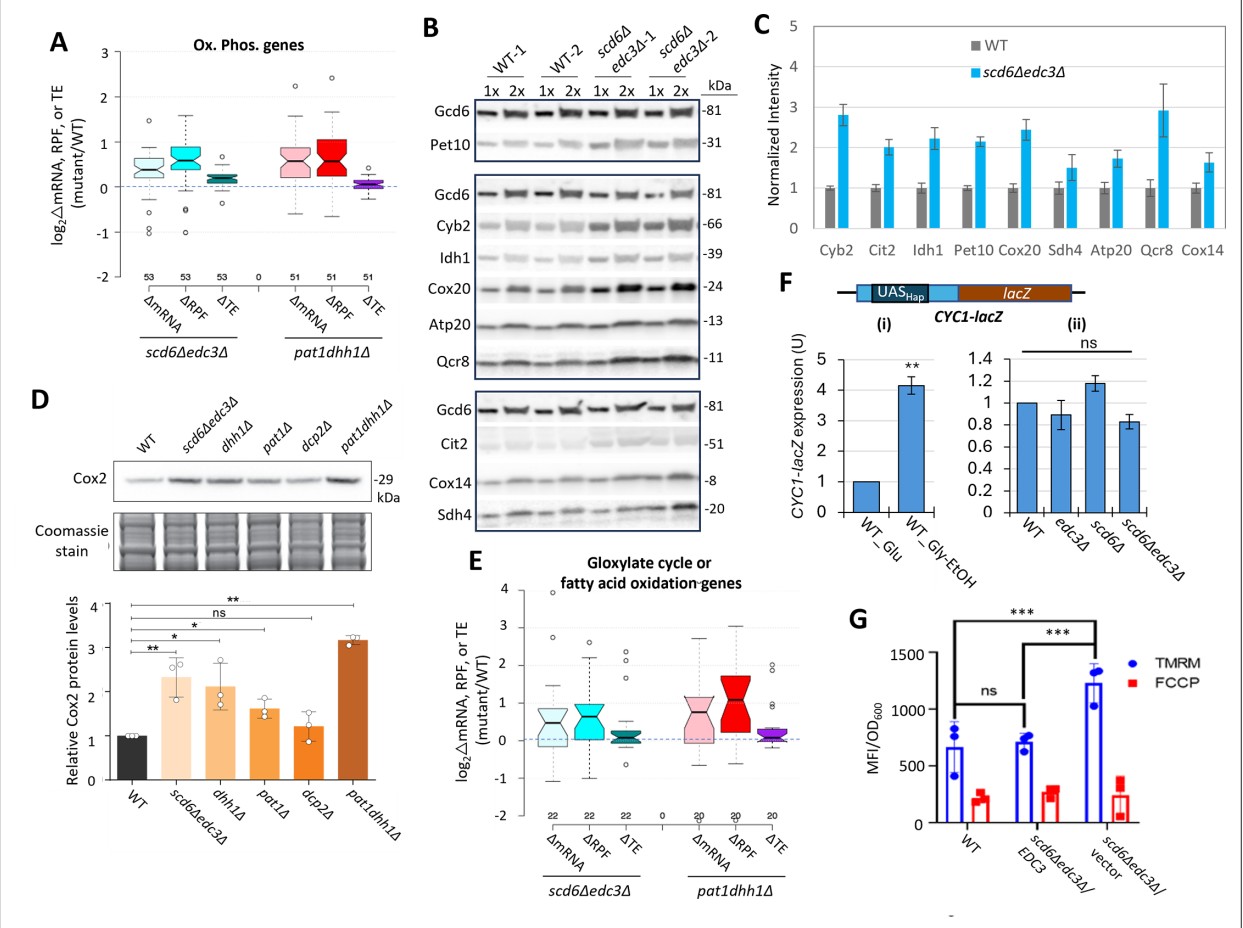

**Figure 6.** Scd6/Edc3 post-transcriptionally repress proteins involved in respiration and suppress mitochondrial membrane potential in rich medium.
(**A**) Log$_2$ changes in mRNA, ribosome-protected fragments (RPFs), or translational efficiency (TE) conferred by the indicated double mutations vs. wild-type (WT) for 53 nuclear genes encoding mitochondrial proteins involved directly in oxidative phosphorylation. (**B–C**) Western blot analysis of nine mitochondrial proteins and Gcd6 (examined as loading control) in WT and *scd6Δedc3Δ* strains, cultured in duplicate in YPD medium to OD$_{600}$ of ~0.6–0.8. WCEs were extracted under denaturing conditions and aliquots corresponding to 1 X or 2 X amounts of WCE were loaded in successive lanes for the two biological replicate cultures. Immune complexes were visualized with enhanced chemiluminescence (**B**). Signals for each protein were quantified, normalized to the corresponding signals for Gcd6 in the same extract and expressed relative to the resulting values for WT cells. Mean values and standard errors are plotted (**C**). (**D**) Western blot analysis of Cox2 in strains of the indicated genotypes in cells cultured as in (**B**). Cox2 signal intensity was normalized to total Coomassie-stained protein and the resulting relative Cox2 protein levels from three biological replicates were averaged and plotted. P-values from student's t-test indicated as **,<0.01; *,<0.05; ns, not significant. (**E**) Log$_2$ changes in mRNA, RPFs, or TE conferred by the indicated double mutations vs. WT for 22 genes encoding enzymes of the glyoxylate cycle or fatty acid metabolism. (**F**) Expression of the *CYC1-lacZ* reporter on plasmid pLG265, lacking UAS1 and containing the optimized version of UAS2, UAS2UP1, in the WT strain grown on SC-Ura medium containing either 2% glucose or 3% glycerol/2% ethanol as carbon sources (i), or in WT and the indicated mutant strains on SC-Ura with 2% glucose (ii). β-galactosidase activity (nmoles of o-nitrophenyl-β-D-galactopyranoside (ONPG) cleaved per min per mg of total protein) was measured in whole cell extracts for three biological replicates of each strain and the mean values were normalized to the mean activity measured in WT grown with glucose as carbon source. **p-value <0.01 from student's t-test; ns, not significant. (**G**) Measurements of mitochondrial membrane potential. WT cells or transformants of the *scd6Δedc3Δ* mutant containing the *EDC3* plasmid pLfz614-7 or empty vector were cultured in SC-Ura to mid-log phase. Tetramethylrhodamine (TMRM) (500 nM) was added and incubated for 30 min before samples were collected and washed once with deionized water. ΔΨ$_m$ was determined by measuring TMRM fluorescence intensity using flow cytometry. Data are presented in arbitrary fluorescence intensity units per OD$_{600}$. Two-way ANOVA was used for statistical analysis and data are given as mean values ± SD (n=3) (****p<0.0001).

The online version of this article includes the following source data and figure supplement(s) for figure 6:

**Source data 1.** Western blot analysis and *lacZ* reporter analysis of *scd6Δedc3Δ* vs. wild-type (WT) strains.

**Source data 2.** Figures of the uncropped blots with the relevant bands labelled used to prepare *Figure 6B*.

**Source data 3.** Original images of the full uncropped, unedited blots used to prepare *Figure 6B*.

**Source data 4.** Figures of the uncropped blots with the relevant bands labelled used to prepare *Figure 6D*.

*Figure 6 continued on next page*

*Figure 6 continued*

**Source data 5.** Original images of the full uncropped, unedited blots used to prepare *Figure 6D*.

**Figure supplement 1.** mRNAs repressed in abundance or translation by Scd6/Edc3 or Dhh1/Pat1 are enriched for common functional categories, including Ox. Phos. proteins and cell wall components.

**Figure supplement 1—source data 1.** Source table for *Figure 6—figure supplement 1*.

**Figure supplement 2.** Scd6/Edc3 post-transcriptionally repress carbon-catabolite-repressed (CCR) genes in rich medium.

**Figure supplement 3.** Synthetic genetic up-regulation of four Dhh1 target mRNAs on combining *scd6Δ* and *edc3Δ* mutations.

**Figure supplement 3—source data 1.** Source table for *Figure 6—figure supplement 3*.

versus glucose as carbon source (*Figure 6—figure supplement 2C(i)*), but was down-regulated by *scd6Δedc3Δ* in glucose-grown cells (panel (ii)).

## Functional evidence that Scd6/Edc3 repress oxidative phosphorylation

We examined the effects of eliminating Scd6/Edc3 on mitochondrial electron transport by measuring mitochondrial membrane potential ($\Delta\Psi_m$) generated by the ETC using the probe tetramethyl-rhodamine (TMRM)—a cationic fluorescent dye that accumulates in mitochondria as a function of $\Delta\Psi_m$. Quantifying dye fluorescence by flow cytometry revealed increased TMRM fluorescence in the *scd6Δedc3Δ* mutant containing an empty vector compared to both the isogenic WT strain and the

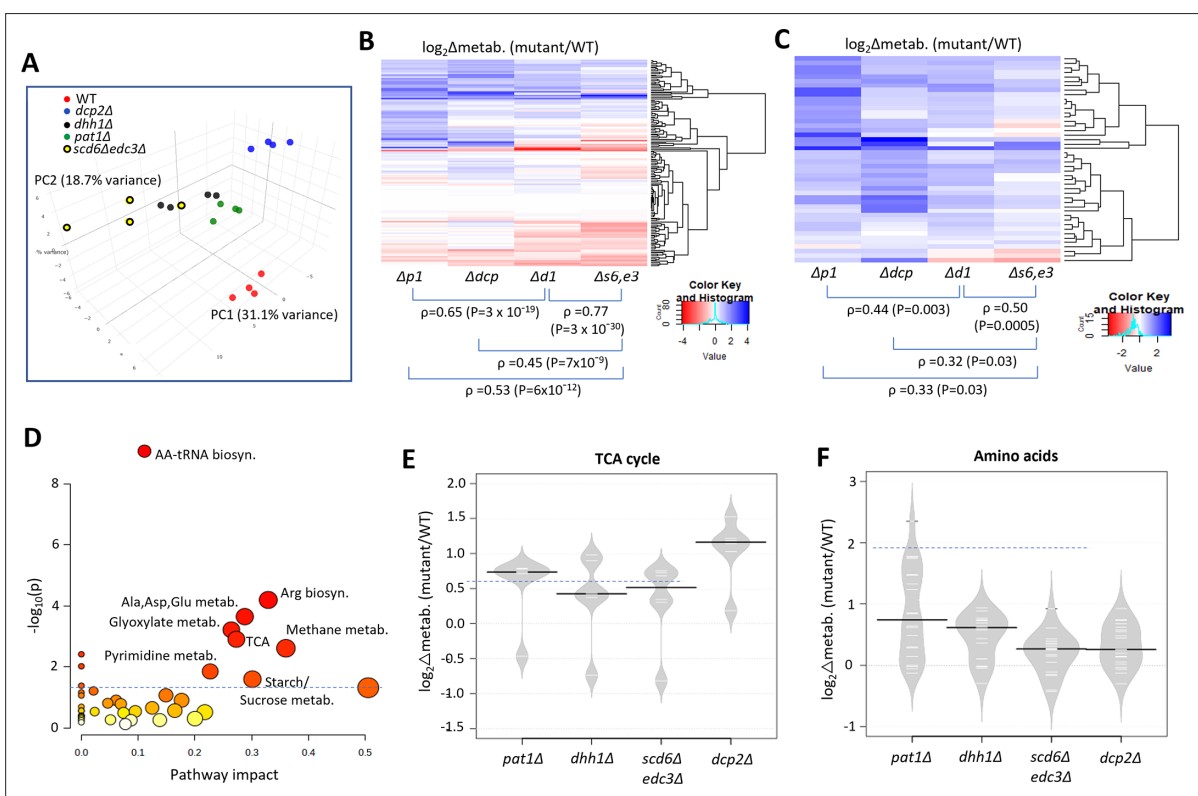

**Figure 7.** Eliminating decapping activators or decapping enzyme confers similar changes in polar metabolites. (**A**) Principal component analysis of the levels of 147 metabolites in biological replicates of each strain. (**B–C**) Hierarchical clustering analysis of log$_2$ changes in all 147 metabolites analyzed (**B**) or the 46 metabolites up-regulated in any two of the four mutants (**C**) conferred by the indicated mutations vs. wild-type (WT), including the Spearman coefficients ($\rho$) and p values for the indicated correlations. (**D**) Results of pathway analysis of the 46 up-regulated metabolites described in (**C**), conducted at https://www.metaboanalyst.ca/MetaboAnalyst/. Red ovals depict groups of metabolites significantly enriched among the set of 46 compounds, with *p*-value <0.05. (**E–F**) Log$_2$ changes in levels of tricarboxylic acid (TCA) cycle intermediates (**E**) or amino acids (**F**) conferred by the indicated mutations vs. WT.

The online version of this article includes the following source data for figure 7:

**Source data 1.** Metabolomics of polar compounds of intermediary metabolism.

mutant complemented by WT *EDC3* (*Figure 6G*). These results are consistent with increased mitochondrial ETC activity in glucose-grown *scd6Δedc3Δ* cells.

To determine whether up-regulation of Ox. Phos. and other glucose-repressed genes in the decapping activator mutants alters cellular metabolites, we used targeted, quantitative LC-MS/MS-based approaches (see Methods) to quantify levels of 147 polar metabolites in the isogenic *scd6Δedc3Δ*, *dhh1Δ*, *pat1Δ*, *dcp2Δ*, and WT strains described above, cultured in YPD medium. Principal component analysis revealed clustering of results from biological replicates in the manner expected for reproducible differences in metabolite levels among different strains, with results for the *scd6Δedc3Δ* mutant most closely resembling those for *dhh1Δ*, which in turn were more similar to the results for *pat1Δ* vs. the *dcp2Δ* mutant or WT strains (*Figure 7A*). This conclusion was borne out by cluster and correlation analyses of changes in metabolites between each mutant compared to WT, with the strongest correlation observed for *scd6Δedc3Δ* vs. *dhh1Δ*, followed by *dhh1Δ* vs. *pat1Δ* (*Figure 7B*). Considering the subset of 46 metabolites up-regulated in any two of the four mutants again showed greatest similarity between changes conferred by *scd6Δedc3Δ* vs. *dhh1Δ* followed by *dhh1Δ* vs. *pat1Δ* (*Figure 7C*). These findings mirror the results from RNA-seq and ribosome profiling, in which the up-regulation of mRNA levels or translation was most similar between the *scd6Δedc3Δ* and *dhh1Δ* mutants. Pathway analysis of the 46 up-regulated metabolites (at https://www.metaboanalyst.ca/MetaboAnalyst/) revealed a significant enrichment for metabolites of both the TCA and glyoxylate cycles (*Figure 7D*), with five of the six TCA cycle intermediates detected (fumarate, malate, α-ketoglutarate, cis-aconitate and citrate) being elevated in the decapping mutants (*Figure 7E*). These results are consistent with the possibility that up-regulation of Ox. Phos. proteins (*Figure 6A–D*) and ETC function (*Figure 6G*) in the decapping mutants leads to increased flux from glucose towards the TCA cycle.

To confirm increased flux through the TCA cycle coming from glucose breakdown, WT and mutant cells were grown in high glucose, pulsed with $^{13}C_6$ glucose, and relative $^{13}C$ label incorporation into TCA cycle intermediates was estimated for each carbon molecule derived from glucose (as indicated schematically in *Figure 8A*) 8 min following the pulse. In all mutants, $^{13}C$ label incorporation in all TCA cycle intermediates was significantly higher than in WT cells (*Figure 8A*), demonstrating increased carbon flux from glucose into the TCA cycle. These results, along with increased ETC activity (*Figure 6G*), indicate that mitochondrial function is up-regulated in these mutants on glucose. If so, we might expect them to show increased ATP production from respiration versus glycolysis. Measuring total ATP levels in glucose-grown cells revealed that, while all decapping mutants except *pat1Δ* showed reduced ATP per cell (*Figure 8B*), the proportion of ATP produced from respiration, and thus eliminated by sodium azide treatment (by inhibiting the ETC), was elevated in all of the mutants (*Figure 8C*).

Interestingly, amino acids are also up-regulated in the decapping mutants (*Figure 7D*), particularly in *pat1Δ* cells (*Figure 7F*), as well as intermediates in amino acid biosynthesis and amino acid derivatives (*Figure 7D*). The expression of amino acid biosynthetic enzymes is unaffected or reduced in the mutants however (*Figure 6—figure supplement 2D*), suggesting that increased amino acid abundance results from metabolic reprogramming rather than increased biosynthetic capacity. One possibility is that accumulation of the TCA cycle intermediate α-ketoglutarate leads to increased production of glutamate and glutamine, precursors in all amino acid biosynthetic pathways (*Ljungdahl and Daignan-Fornier, 2012*). The two 3-phosphotrioses generated in glycolysis (glyceraldehyde 3-phosphate and dihydroxyacetone phosphate) are also elevated in the mutants, possibly owing to increased glyoxylate shunt function, which might stimulate the synthesis of amino acids serine and glycine (*Ljungdahl and Daignan-Fornier, 2012*; *Figure 7D*, Methane metabolism category). The increases in fructose 1,6-bisphosphate, fructose 6-phosphate, glucose 6-phosphate, and UDP-glucose in all four mutants (*Figure 7D*, Starch/Sucrose metabolism category) all suggest substantial metabolic rewiring indicative of glucose up-regulation and usage of alternative carbon sources. In addition to amino acids, pyrimidine nucleotides are up-regulated in the mutants (*Figure 7D*, Pyrimidine metab.), which might be driven by the elevated glutamine levels or increased glycolytic flux towards the pentose phosphate pathway (*Ljungdahl and Daignan-Fornier, 2012*).

## Discussion

RNA-seq analysis of single and double *scd6Δ* and *edc3Δ* mutants has revealed that Scd6 and Edc3 have largely overlapping functions in repressing mRNA abundance, as the presence of either protein alone is sufficient for nearly WT levels of most mRNAs found up-regulated in the double mutant

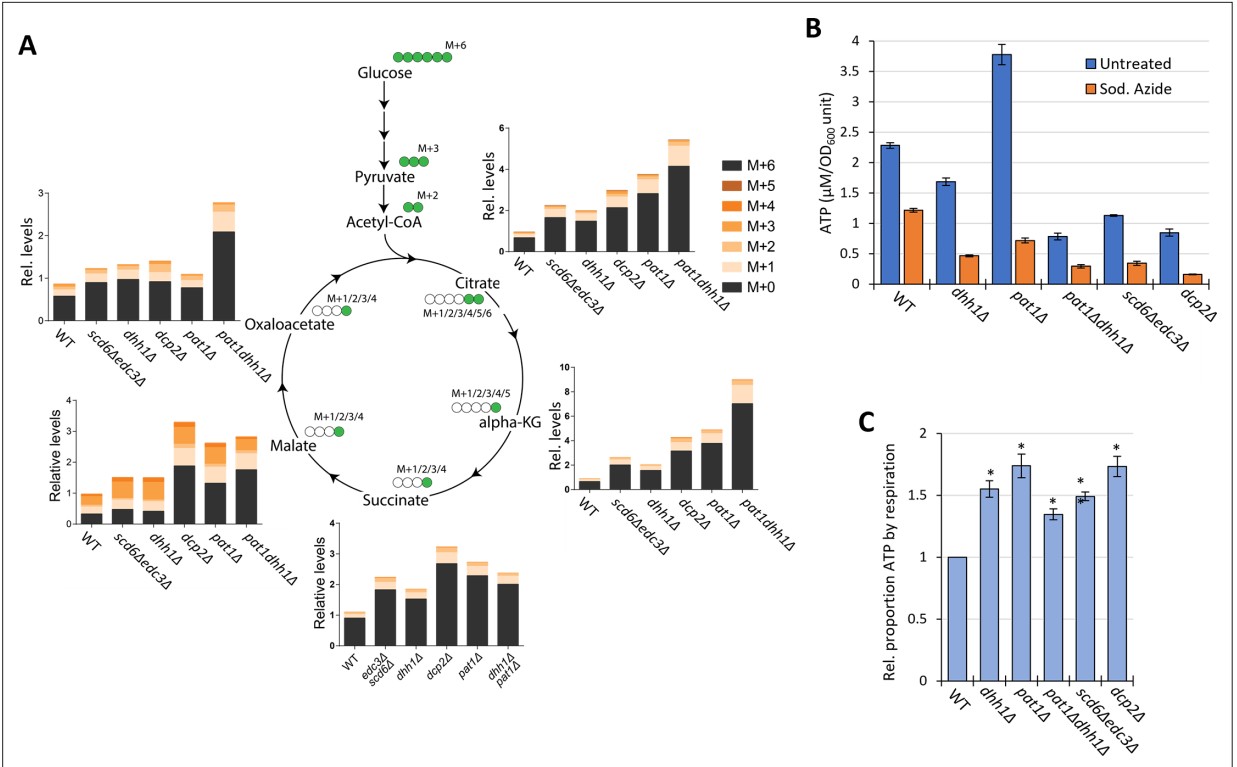

**Figure 8.** Elimination of decapping activators or decapping enzyme up-regulates respiration, increasing flux from glucose into TCA cycle intermediates and proportion of ATP produced by Ox. Phos. (**A**) Three biological replicates of cells of each indicated genotype were cultured in YP with 2% unlabeled glucose, shifted to YP with 1% unlabeled glucose for 20 min, and pulsed with $^{13}C_6$ labeled glucose (at final concentration of 1%) for 8 min, followed by extraction of metabolites and quantification of the indicated TCA cycle intermediates by mass spectrometry. In the diagrams, green circles signify the labeled carbon atom and the notation M+1, M+2, etc., indicates the mass increase in the molecules due to the labeled carbon. The depicted labeling pattern of metabolites reflects one cycle of the TCA cycle, resulting in mass additions of M+1 and M+2; however, across multiple cycles, a broader range of metabolite species with different mass additions will emerge. The metabolite signal intensities in all samples are expressed relative to that determined for the first replicate of the wild-type (WT) strain. (**B–C**) Measurements of ATP levels and proportions of total ATP impaired by azide inhibition of ETC activity. Cells cultured in YPD were treated or untreated with sodium azide for 30 min prior to harvesting. ATP levels were determined in extracts and normalized to $OD_{600}$ units of cells for three biological replicates each of treated and untreated cell aliquots. Mean values for each strain are plotted in (**B**) and relative fractions of ATP in untreated samples retained following azide treatment (ATP_untreated)-(ATP_Azide)/(ATP_untreated), normalized to the values determined for WT are plotted in (**C**) with results from a student's t-test indicated with asterisks: **$p<0.005$; *$p<0.05$.

The online version of this article includes the following source data and figure supplement(s) for figure 8:

**Source data 1.** Glucose flux analysis.

**Source data 2.** Measurements of cellular ATP content.

**Figure supplement 1.** Hypothetical model to explain concerted repression of mRNA abundance or translation of particular mRNAs by decapping activators Scd6, Edc3, Dhh1, and Pat1.

(*Figure 1—figure supplement 3A* and *Figure 1B–C(ii–iii)*). Although the mRNAs dysregulated in the *scd6Δedc3Δ* mutant are enriched for ESR mRNAs; the majority are not ESR transcripts (*Figure 1A–B*) and are most likely repressed more directly by Scd6/Edc3. Functional redundancy of Scd6/Edc3 is consistent with their similarities in sequence and domain structure, including shared FDF motifs that interact competitively with Dhh1 in animals (*Tritschler et al., 2008*; *Tritschler et al., 2009*) and N-terminal LSm domains that compete for binding motifs in the Dcp2 CTT of *S. pombe* (*Fromm et al., 2012*), as well as the synthetic growth defect produced by deleting both genes simultaneously (*Decourty et al., 2008*; *Figure 1—figure supplement 1A*). Interestingly, a set of 37 mRNAs appears to be repressed by Edc3 exclusively (*Figure 1B–C(i)*), including the canonical Edc3 targets identified previously, *YRA1* and *RPS28B* (*He et al., 2022*), and is enriched for genes belonging to the GO category 'mitochondrion' (22/37 genes) and the Ox. Phos. categories of electron transport and mitochondrial ATP synthesis (11/37 genes: *ATP3 ATP16 ATP5 TIM11 ATP17 COX4 QCR9 CYC1 COX12 ATP18 COX5A*). The fact that deleting *SCD6* in the double mutant frequently diminishes the up-regulation

of these transcripts conferred by deleting *EDC3* alone (*Figure 1C(i)*) might indicate that eliminating both Scd6 and Edc3 together enables a distinct degradation pathway that compensates for loss of Edc3-stimulated decapping/decay.

Several lines of evidence support the conclusion that most of the changes in mRNA abundance in the *scd6Δedc3Δ* double mutant result from impaired decapping and attendant 5'–3' degradation by Xrn1. ChIP-seq analysis of Rpb1 shows that Pol II occupancies in the coding sequences increase by much smaller amounts compared to the increased transcript levels for non-iESR mRNAs up-regulated in *scd6Δ/edc3Δ* cells (*Figure 3E*), indicating a minor contribution of increased transcription to their up-regulation. The transcriptional activators of Ox-Phos. and CCR genes, many of whose transcripts are up-regulated by *scd6Δ/edc3Δ*, remain largely inert in this mutant, ruling out inappropriate transcriptional induction of their target genes in glucose-replete *scd6Δ/edc3Δ* cells. Consistent with mRNA turnover via decapping, most of these transcripts are up-regulated similarly by *scd6Δedc3Δ* and deletion of *DCP2* (*Figure 3A–B*), exhibit a heightened proportion of decapped isoforms in WT cells that is diminished by deleting *SCD6/EDC3* or *DHH1* (*Figure 3C* and *Figure 1—figure supplement 3E*), and show evidence of co-translational 5'–3' decay of decapped intermediates by Xrn1 (*Figure 3D*); none of which was observed for the mRNAs down-regulated in *scd6Δ/edc3Δ* cells. Similar findings were reported previously for sets of mRNAs up-regulated by *dhh1Δ* or *pat1Δ* (*Vijjamarri et al., 2023a*), consistent with the widespread involvement of Dhh1 and Pat1 in controlling the levels of mRNAs preferentially targeted by Scd6/Edc3.

Supporting this last assertion, most (~70%) of the mRNAs repressed in abundance by Edc3/Scd6 are also repressed by Dhh1 and Pat1, and their efficient repression in WT cells involves independent contributions of similar magnitude by Pat1, Dhh1, and either Edc3 or Scd6 (*Figure 2A–B(ii)*). Another large set of mRNAs is repressed predominantly by Pat1 with lesser contributions by Dhh1 and Scd6/Edc3 (*Figure 2A–B(iii)*). Our finding of extensive cooperation among these four decapping factors is consistent with our recent finding that ~55% of the mRNAs up-regulated in the *dcp2Δ* mutant, and thus targeted by Dcp2 for enhanced degradation, tend to be up-regulated in the *pat1Δdhh1Δ* and *scd6Δedc3Δ* mutants, whereas the remaining 45% are generally up-regulated by *upf1Δ* instead. This finding suggested a major bifurcation of Dcp2 activation by either Pat1/Dhh1/Scd6/Edc3 or the Upf factors responsible for NMD (*Vijjamarri et al., 2023b*).

Closer examination of the effects of individual mutations on mRNA levels revealed a greater overlap between the mRNAs dysregulated by *dhh1Δ* and *scd6Δ/edc3Δ* versus those altered by *pat1Δ* (*Figure 2C–E*), consistent with the aforementioned FDF motifs in Scd6 and Edc3 that interact with Dhh1 in animal systems, and with evidence for distinct complexes of the decapping enzyme containing Dhh1 and Edc3 or Scd6 but lacking Pat1. It also supports the recent suggestion that Edc3 and Scd6 act interchangeably to recruit Dhh1 to the Edc3-interaction site in the Dcp2 CTT to stimulate turnover of several Dhh1 target mRNAs (*He et al., 2022*). Our RNA-seq results support this last proposal in showing that all four Dhh1 target mRNAs examined in that study (*EDC1, SDS23, HXT6,* and *HSP12*) are up-regulated in the *scd6Δedc3Δ* double mutant while changing little in the single mutants (*Figure 6—figure supplement 3*), two of which (*SDS23* and *HXT6*) were shown previously to have longer half-lives in *dhh1Δ* vs. WT cells (*He et al., 2018*). Finding that selective up-regulation in the *scd6Δedc3Δ* double mutant applies to most mRNAs up-regulated in *dhh1Δ* cells (*Figure 2E*), we propose that redundant Scd6/Edc3 targeting is a widespread mechanism of Dhh1-enhanced mRNA degradation in the yeast transcriptome (*Figure 8—figure supplement 1*). Supporting this, our co-immunoprecipitation analysis revealed diminished Dhh1-Dcp2 association in cell extracts of the *scd6Δedc3Δ* double mutant but not for either single mutant (*Figure 3F*).

Examining our Ribo-Seq and TMT-MS data suggests that *DHH1* mRNA translation and steady-state abundance are increased ~ twofold in the *scd6Δedc3Δ* strain, indicating that up-regulation of many of the same mRNAs by *scd6Δedc3Δ* and *dhh1Δ* does not result indirectly from reduced levels of Dhh1 in the *scd6Δedc3Δ* mutant. The increased Dhh1 expression might signify a compensatory response to the absence of Scd6/Edc3. We also observed an ~40% reduction in Dcp2 translation and abundance in the *scd6Δedc3Δ* strain, which might contribute to the up-regulation of mRNAs dysregulated in this mutant. However, our immunoblot analyses revealed no significant reduction in steady-state Dcp2 levels in *scd6Δedc3Δ* cells (Input lanes in *Figure 3F*, *Figure 3—figure supplement 2(i–ii)*). Moreover, our previous finding that the majority of mRNAs subject to NMD, up-regulated by both *upf1Δ* and *dcp2Δ*, are not up-regulated by *scd6Δedc3Δ* (*Vijjamarri et al., 2023b*) implies that Dcp2 abundance

in *scd6Δedc3Δ* cells is adequate for normal levels of NMD. Accordingly, we favor a direct role for Scd6/Edc3 in accelerating degradation of most transcripts up-regulated in the *scd6Δedc3Δ* mutant.

Despite strong evidence for the involvement of decapping in mRNA turnover directed by either Scd6/Edc3 or Pat1/Dhh1, it appears that neither enrichment for non-optimal codons nor low translational efficiencies are principal determinants of preferential targeting of mRNAs by these proteins (*Figure 3—figure supplement 3A–C*). There are probably exceptions to these generalizations among the hundreds of transcripts repressed by these factors; nor can we rule out the involvement of clusters of non-optimal codons in transcripts of average overall codon optimality that generate a queue of slowly elongating ribosomes that triggers decapping. Nevertheless, it seems likely that other sequences or properties of the transcripts most strongly repressed by Scd6/Edc3 or Dhh1/Pat1 frequently underlie their preferential targeting by these factors for decapping/decay. Sequence-specific RNA-binding proteins that actively recruit these decapping activators is an attractive potential mechanism (*Figure 8—figure supplement 1*). For example, there is evidence that Dhh1 is recruited to mRNAs in association with the Ccr4-NOT deadenylase via RNA-binding protein Puf5 (*Goldstrohm et al., 2006*). Involvement of RNA-binding proteins would also be consistent with the observed specific targeting of mRNAs whose expression is repressed in response to nutrient abundance.

Ribosome profiling of the single and double *scd6Δ* and *edc3Δ* mutants provided evidence that Scd6 and Edc3 also have highly overlapping functions in repressing the translation of ~200 mRNAs, whose TE values are up-regulated substantially only in the double mutant (*Figure 4A–B(iii)*). The observation that changes in TE (RPF density) generally correlate with changes in protein abundance conferred by *scd6Δedc3Δ* (*Figure 4F*) implies that Scd6/Edc3 generally repress translation at the stage of initiation rather than elongation. Many mRNAs translationally repressed by Scd6/Edc3 additionally require Dhh1, and the subset most strongly repressed also requires Pat1 for efficient translational repression (*Figure 5B(ii) and D*). This last group of mRNAs is translated at very low levels in WT cells on rich medium (*Figure 5—figure supplement 1C*, cols. 2 & 4), which might reflect the concerted action of all four decapping activators in suppressing translation. The transcript abundance of these mRNAs is also up-regulated in each of the mutants (*Figure 5—figure supplement 1A(ii)*), indicating enhanced degradation coupled with translational repression by the four decapping activators. One possibility is that a complex of Dcp1:Dcp2 containing Scd6/Edc3, Pat1/Dhh1, or different combinations of these factors (*He et al., 2022*) is recruited to these mRNAs and impedes recruitment of PICs to diminish translation in parallel with stimulating decapping and 5'–3' degradation by Xrn1 (*Figure 8—figure supplement 1*).

The additional sets of mRNAs identified here that are translationally repressed exclusively by Scd6/Edc3 or Pat1/Dhh1 (*Figure 5A(i) and (iii)*) show much less evidence of transcript degradation promoted by these factors (*Figure 5—figure supplement 1A(i) and (iii)*), suggesting that inhibition of translation initiation occurs without decapping or involves decapping uncoupled from degradation by Xrn1 (*Figure 8—figure supplement 1(ii–iii)*). In the latter scenario, the decapped mRNAs would be unable to recruit the cap-binding initiation factors necessary for PIC recruitment and hence persist in WT cells as translationally inert degradation intermediates. Eliminating the decapping activators would diminish decapping and eliminate these fractions of poorly translated uncapped isoforms to confer the increased TE values we observed for the cognate genes in the decapping mutants. This hypothetical mechanism of translational repression via decapping is consistent with the lower than average capped to total mRNA (C/T) ratios found in WT cells for the mRNAs translationally repressed by all four decapping activators or by Pat1/Dhh1 (*Figure 5—figure supplement 1D(ii–iii)*), indicating accumulation of decapped isoforms in WT cells, as observed previously for a subset of mRNAs translationally repressed by Dhh1 (*Vijjamarri et al., 2023b*). In contrast, the mRNAs that are translationally repressed exclusively by Scd6/Edc3 display higher than average C/T ratios in WT (*Figure 5—figure supplement 1D(i)*), which is more compatible with an inhibition of PIC recruitment independently of decapping (*Figure 8—figure supplement 1(iii)*). Translational repression without decapping could also occur if transcripts bound by decapping complexes are sequestered from the translation initiation machinery in RNA granules where mRNA decapping occurs inefficiently (*Figure 8—figure supplement 1(iv)*). Finally, it is important to note that mRNA decay occurs apparently without translational control for the much larger groups of ~600 and ~1000 transcripts showing up-regulated mRNA abundance but little evidence of TE changes in the *scd6Δedc3Δ* and *pat1Δdhh1Δ* mutants, respectively (*Figure 5—figure supplement 1B*). It is unclear whether these mRNAs escape translational repression

and are targeted exclusively for decapping/degradation (*Figure 8—figure supplement 1(i)*) or are simply degraded too fast to allow translational repression to be detected by ribosomal profiling of steady-state mRNAs.

Recently, we reported that Pat1 and Dhh1 functionally collaborate with the decapping enzyme to repress genes and cellular pathways normally repressed in rich medium (*Vijjamarri et al., 2023a*; *Vijjamarri et al., 2023b*). Here, we found evidence that Scd6 and Edc3 broadly cooperate with Pat1/Dhh1 in this control, acting redundantly to repress the abundance, translation, or both, of mRNAs encoding numerous proteins involved in respiration or utilization of alternative carbon sources (*Figure 6A–C*, *Figure 6—figure supplement 1A–D*), without affecting transcription of the cognate genes (*Figure 6F*, *Figure 6—figure supplement 2A–C*). Evidence that up-regulation of Ox. Phos. mRNAs/proteins in the *scd6Δedc3Δ* mutant is sufficient to increase respiration in glucose-replete cells came from our findings of increased mitochondrially-encoded subunit II of ETC component cytochrome *c* oxidase (*Figure 6D*) and elevated mitochondrial membrane potential (*Figure 6G*) in the *scd6Δedc3Δ* mutant, extending similar findings we made previously for *dcp2Δ*, *dhh1Δ*, and *pat1Δ* mutants (*Vijjamarri et al., 2023a*; *Vijjamarri et al., 2023b*). Consistent with this, our metabolomics analysis revealed elevated steady-state levels of TCA cycle intermediates in all four decapping mutants (*Figure 7D*), and increased $^{13}$C carbon incorporation from $^{13}C_6$-glucose breakdown into TCA cycle intermediates in the same mutant strains (*Figure 8A*). Analysis of ATP levels further revealed increased proportions of ATP derived from Ox. Phos. in the decapping mutants (*Figure 8C*), which together provide strong evidence that mitochondrial respiration is inappropriately elevated in glucose-replete cells of decapping activator mutants. It is well established that the enzymes involved in respiration and catabolism of non-glucose carbon sources are repressed at the transcriptional level in glucose-replete yeast (*Zaman et al., 2008*). Our findings demonstrate that Scd6/Edc3 collaborate with Dhh1/Pat1 to provide an additional layer of post-transcriptional control of these enzymes through enhanced decapping/decay or translational repression to help ensure their complete suppression when glucose is abundant (*Figure 8—figure supplement 1*).

# Methods

## Key resources table

| Reagent type (species) or resource | Designation | Source or reference | Identifiers | Additional information |
|---|---|---|---|---|
| Chemical compound, drug | Cycloheximide | Sigma | Cat # C7698 | |
| Chemical compound, drug | 5-fluoro-orotic acid | US Biological | Cat # F5050 | |
| Chemical compound, drug | ONPG (o-nitrophenyl-β-D-galactopyranoside) | Sigma | Cat # N1127 | |
| Chemical compound, drug | TCA (Trichloroacetic acid) solution | Sigma | Cat # T0699 | |
| Chemical compound, drug | G418 Sulfate (Geneticin) | US Biological | Cat # G1000 | |
| Chemical compound, drug | IgG sepharose | Cytiva | Cat # 17-0969-01 | |
| Commercial assay or kit | NEBuilder HiFi DNA assembly | New England Biolabs | Cat # E2621S | |
| Commercial assay or kit | RNase I | Ambion | Cat # AM2294 | |
| Commercial assay or kit | QIAzol Lysis reagent | Qiagen | Cat. # 79306 | |
| Commercial assay or kit | miRNeasy Mini Kit | Qiagen | Cat. # 217004 | |
| Commercial assay or kit | RNA Clean and Concentrator kit | Zymo Research | Cat # R1018 | |
| Commercial assay or kit | T4 Polynucleotide kinase | New England Biolabs | Cat # M0201L | |
| Commercial assay or kit | T4 Rnl2(tr) K227Q | New England Biolabs | Cat # M0351S | |
| Commercial assay or kit | 5′ deadenylase/RecJ exonuclease | Epicentre | Cat # RJ411250 | |
| Commercial assay or kit | Oligo Clean and Concentrator column | Zymo Research | Cat # D4060 | |
| Commercial assay or kit | Protoscript II | New England Biolabs | Cat # M0368L | |
| Commercial assay or kit | CircLigase ssDNA Ligase | Epicenter | Cat # CL4111K | |
| Commercial assay or kit | Phusion polymerase | New England Biolabs | Cat # M0530S | |

*Continued on next page*

*Continued*

| Reagent type (species) or resource | Designation | Source or reference | Identifiers | Additional information |
|---|---|---|---|---|
| Commercial assay or kit | High Sensitivity DNA Kit | Agilent | Cat # 5067–4,626 | |
| Commercial assay or kit | Fragmentation Reagent | Ambion | Cat # AM8740 | |
| Commercial assay or kit | Stop Solution | Ambion | Cat # AM8740 | |
| Commercial assay or kit | Ribo-Zero Gold rRNA Removal Kit | Illumina | Cat # MRZ11124C | |
| Commercial assay or kit | RNA 6000 Nano kit | Agilent | Cat # 5067–1511 | |
| Commercial assay or kit | ERCC ExFold RNA spike-In Mixes | Ambion | Cat # 4456739 | |
| Commercial assay or kit | DNA Library Prep Kit | New England Biolabs | Cat # E7370L | |
| Commercial assay or kit | DNase I | Roche | Cat # 4716728001 | |
| Commercial assay or kit | Luciferase Control RNA | Promega | Cat # L4561 | |
| Commercial assay or kit | Superscript III First-Strand synthesis kit | Invitrogen | Cat # 18080051 | |
| Commercial assay or kit | 25 mM triethylammonium-bicarbonate | Thermo Scientific | Cat # 90114 | |
| Commercial assay or kit | GelCode Blue Stain | Thermo Scientific | Cat # 24592 | |
| Commercial assay or kit | Pierce BCA Protein Assay Kit | Thermo Scientific | Cat # 23225 | |
| Commercial assay or kit | protease inhibitor cocktail | Roche | Cat # 5056489001 | |
| Commercial assay or kit | enhanced chemiluminescence (ECL) | Cytiva | Cat # RPN2109 | |
| Commercial assay or kit | Nano-Glo substrate | Promega | Cat # N1120 | |
| Commercial assay or kit | Bradford reagent | BioRad | Cat # 5000006 | |
| Commercial assay or kit | ATP Determination Kit | Thermo-Fisher Scientific | Cat # A22066 | |
| Commercial assay or kit | In-Fusion Snap Assembly Master Mix | Takara | Cat # 638949 | |
| Antibody | Anti-Gcd6, rabbit polyclonal | In house; *Bushman et al., 1993* | | Dilution: 1:2,000 (WB) |
| Antibody | Anti-Pet10, rabbit polyclonal | Dr. Nikolaus Pfanner | | Dilution: 1:1,000 (WB) |
| Antibody | Anti-Cyb2, rabbit polyclonal | Dr. Thomas Fox | | Dilution: 1:2,000 (WB) |
| Antibody | anti-Idh1, goat polyclonal | Abnova | Cat # PAB19472 | Dilution: 1:500 (WB) |
| Antibody | Anti-Cox20, rabbit polyclonal | Dr. Thomas Fox | | Dilution: 1:500 (WB) |
| Antibody | Anti-Atp20, rabbit polyclonal | Dr. Nikolaus Pfanner | | Dilution: 1:500 (WB) |
| Antibody | Anti-Qcr8, rabbit polyclonal | Dr. Nikolaus Pfanner | | Dilution: 1:500 (WB) |
| Antibody | anti-Cit2, rabbit polyclonal | https://www.antibodies-online.com/ | Cat # ABIN4889057 | Dilution: 1:250 (WB) |
| Antibody | Anti-Cox14, rabbit polyclonal | Dr. Nikolaus Pfanner | | Dilution: 1:500 (WB) |
| Antibody | Anti-Sdh4, rabbit polyclonal | Dr. Nikolaus Pfanner | | Dilution: 1:500 (WB) |
| Antibody | anti-TAP, rabbit polyclonal | Thermo-Fisher Scientific | Cat # CAB1001 | Dilution: 1:2000 (WB) |
| Antibody | anti-HA (12CA5) mouse monoclonal | Roche | Cat # 10522600 | Dilution: 1:4000 (WB) |
| Antibody | anti-mouse IgG (HRP-conjugated), mouse polyclonal | GE HealthCare | Cat # NA931V | Dilution: 1:10000 (WB) |
| Antibody | anti-goat IgG (HRP-conjugated), chicken polyclonal | Abnova | Cat # PAB29101 | Dilution: 1:10000 (WB) |
| Antibody | anti-Rpb1 8WG16, mouse monoclonal | Biolegend | Cat # 664906; RRID:AB_2565554 | 4 µl used in Rpb1-ChIP |
| Strain, strain background (*Saccharomyces cerevisiae*) | Yeast strains used | This paper | RRID:NCBITaxon_2305205 | *Table 1* |
| Genetic reagent (plasmid) | Plasmids used | This paper | | *Table 2* |

*Continued on next page*

*Continued*

| Reagent type (species) or resource | Designation | Source or reference | Identifiers | Additional information |
|---|---|---|---|---|
| Sequence-based reagent | Primers used | This paper | | *Table 3* |
| Software, algorithm | Notched box-plots | http://shiny.chemgrid.org/boxplotr/ | RRID:SCR_015629 | |
| Software, algorithm | Venn diagrams | https://www.biovenn.nl/ | RRID:SCR_026853 | |
| Software, algorithm | Hypergeometric distribution | https://systems.crump.ucla.edu/hypergeometric/index.php | | |
| Software, algorithm | Volcano plots | https://huygens.science.uva.nl/VolcaNoseR/ | RRID:SCR_025419 | |
| Software, algorithm | Gene ontology (GO) | http://funspec.med.utoronto.ca/ | RRID:SCR_006952 | |
| Software, algorithm | DESeq2 analysis | *Zhang, 2023* | | |
| Software, algorithm | Integrative Genomics Viewer | http://software.broadinstitute.org/software/igv/ | RRID:SCR_011793 | IGV 2.4.14 |
| Software, algorithm | Genome-wide occupancy profiles for Rpb1 (ChIP) | *Zhang, 2022* | | |
| Software, algorithm | Image Lab 6.0.1 program | https://www.bio-rad.com/en-us/product/image-lab-software?ID=KRE6P5E8Z | RRID:SCR_014210 | |
| Software, algorithm | SwissProt Yeast database | https://www.uniprot.org/proteomes/UP000002311 | | |
| Software, algorithm | Proteome Discoverer 2.4 | Thermo SCR_002798 | RRID:SCR_014477 | |
| Software, algorithm | GraphPad Prism 9 | GraphPad Software, San Diego, CA | RRID:SCR_002798 | 9.4.1 |

**Table 1.** Yeast strains employed.

| Strain | Genotype | Source |
|---|---|---|
| 255 | *MATa his3Δ1 leu2Δ0 met15Δ0 ura3Δ0 edc3Δ::kanMX4* | Research Genetics |
| HFY114 (W303) | *MATa ade2-1 ura3-1 his3-11,15 trp1-1 leu2-3, 112 can1-100* | *He et al., 2003* |
| SYY2352 | *MATa ade2-1 ura3-1 his3-11,15 trp1-1 leu2-3, 112 can1-100 scd6Δ::kanMX6* | *He and Jacobson, 2015* |
| FZY855 | *MATa ade2-1 ura3-1 his3-11,15 trp1-1 leu2-3, 112 can1-100 scd6Δ::hphMX4* | This study |
| FZY858 | *MATa ade2-1 ura3-1 his3-11,15 trp1-1 leu2-3, 112 can1-100 scd6Δ::hphMX4 edc3Δ::kanMX4* | This study |
| FZY862 | *MATa ade2-1 ura3-1 his3-11,15 trp1-1 leu2-3, 112 can1-100 edc3Δ::kanMX4* | This study |
| H5217/QZY126 | *MATa ade2-1 ura3-1 his3-11,15 trp1-1 leu2-3,112 can1-100 dhh1Δ::kanMX* | *Zeidan et al., 2018* |
| F2181/BSY3037 | *MATa ade2-1 ura3-1 his3-11,15 trp1-1 leu2-3,112 can1-100 pat1Δ::HIS3* | *Charenton et al., 2017* |
| F2182/YFW168 | *MATa ade2-1 ura3-1 his3-11,15 trp1-1 leu2-3,112 can1-100 pat1Δ::HIS3 dhh1Δ::kanMX* | *Charenton et al., 2017* |
| CFY1016 | *MATa ade2-1 ura3-1 his3-11,15 trp1-1 leu2-3,112 can1-100 dcp2::HIS3* | *He et al., 2003* |
| F2262 | *MATa his3-Δ1 leu2-Δ0 met15-Δ0 ura3-Δ0 DHH1::TAP::HIS3MX* | GE Healthcare Dharmacon |
| H5695 | *MATa ade2-1 ura3-1 his3-11,15 leu2-3,112 can1-100 DHH1- TAP::HIS3MX* | This study |
| H5696 | *MATa ade2-1 ura3-1 his3-11,15 leu2-3,112 can1-100 edc3Δ::kanMX4 DHH1-TAP::HIS3MX* | This study |
| H5697 | *MATa ade2-1 ura3-1 his3-11,15 leu2-3,112 can1-100 scd6Δ::kanMX4 DHH1-TAP::HIS3MX* | This study |
| H5698 | *MATa ade2-1 ura3-1 his3-11,15 leu2-3,112 can1-100 scd6Δ::hphMX4 edc3Δ::kanMX4 DHH1-TAP::HIS3MX* | This study |

## Yeast strains and plasmids

Unless otherwise indicated, the following strains of yeast *Saccharomyces cerevisiae* were employed for all experiments: WT strain HFY114 (W303: *MATa ade2-1 ura3-1 his3-11,15 trp1-1 leu2-3,112 can1-100*) (*He et al., 2003*), *scd6Δ* strain SYY2352 (*MATa ade2-1 ura3-1 his3-11,15 trp1-1 leu2-3,112 can1-100 scd6Δ::kanMX6*) (*He and Jacobson, 2015*), *edc3Δ* strain FZY862 (*MATa ade2-1 ura3-1 his3-11,15 trp1-1 leu2-3,112 can1-100 edc3Δ::kanMX4*), and *scd6Δedc3Δ* strain FZY858 (*MATa ade2-1 ura3-1 his3-11,15 trp1-1 leu2-3,112 can1-100 scd6Δ::hphMX4 edc3Δ::kanMX4*).

Strains FZY858 and FZY862 were generated by replacing chromosomal *EDC3* in strains FZY855 and HFY114, respectively, with the *edc3Δ::kanMX4* allele amplified by PCR from the chromosomal DNA of strain 255 (*MATa his3Δ1 leu2Δ0 met15Δ0 ura3Δ0 edc3Δ::kanMX4*) obtained from Research Genetics. DNA sequences up to 310 bp upstream and 340 bp downstream of the *EDC3* coding sequence were included in the amplified fragment used for transformation of FZY855 and HFY114 to G418-resistance. FZY855 was derived from SYY2352 by transforming with marker-swap plasmid pAG32 (*Goldstein and McCusker, 1999*) to replace *scd6Δ::kanMX6* with *scd6Δ::hphMX6,* selecting for resistance to hygromycin and screening for loss of G418-resistance. The presence of all deletion alleles was verified by PCR analysis of chromosomal DNA.

Strains H5695, H5696, H5697, and H5698 strains, employed for co-immunoprecipitation experiments, were generated by PCR amplification of a *DHH1-TAP-HIS3MX* fragment from strain F2262 using primers AKV224 and AKV225, containing 474 bp upstream and 450 bp downstream of the *DHH1* stop codon, that was used to transform strains HFY114, FZY862, SYY2352, and FZY858, respectively, to His⁺. A complete list of yeast strains employed is given in *Table 1* and all strains are available on request.

Plasmids pLfz614-7 and pLfz635-5 contain the *EDC3* CDS with 420 bp upstream and 490 bp downstream, on a ~2.6 kb fragment, and plasmids pL615-5 and pLfz636-1 contain the *SCD6* CDS with 500 bp upstream and 180 bp downstream on a ~1.7 kb fragment, amplified by PCR from yeast genomic DNA and inserted between the XhoI and EcoRI sites of YCplac33 or YCplac111, respectively. Plasmid pAK133 containing *DCP2-3HA* was produced by a multi-step PCR amplification scheme. First, the *DCP2* CDS with 920 nucleotides upstream of the start codon and 587 nucleotides downstream of the stop codon was amplified from genomic DNA of WT strain HFY114. Sequences encoding the 3XHA tag were introduced immediately upstream of the *DCP2* stop codon using primer pairs AKV372/AKV373 and AKV374/AKV375 during the first round of PCR amplification; a second round of PCR followed using primers AKV372 and AKV375 to amplify a complete *DCP2-3HA* fragment also containing sequences homologous to the ends of plasmid YCplac33 linearized by digestion with BamHI. The PCR product was fused into BamHI-digested YCplac33 using the Takara In-Fusion cloning kit (#638956) according to the manufacturer's instructions. Primers AKV372 and AKV375 were designed using the Takara online primer design tool to ensure optimal homologous recombination during the cloning process. The inserted yeast DNA fragments were verified by sequencing in their

**Table 2.** Plasmids employed.

| Plasmid | Description | Source |
|---|---|---|
| YCplac33 | s.c. *URA3* vector | *Gietz and Sugino, 1988* |
| YCplac111 | s.c. *LEU2* vector | *Gietz and Sugino, 1988* |
| pLfz614-7 | *EDC3* in YCplac33 | This study |
| pLfz615-5 | *SCD6* in YCplac33 | This study |
| pLfz635-5 | *EDC3* in YCplac111 | This study (in case used) |
| pLfz636-1 | *SCD6* in YCplac111 | This study (in case used) |
| pLGADH2 | *ADH2* 5' non-coding region fused to *lacZ* | *Sloan et al., 1999* |
| pLG265 | *CYC1-lacZ* reporter lacking UAS1 and containing the optimized version of UAS2, UAS2UP1 | *Forsburg and Guarente, 1989* |
| pRK4 | UAS$_{GATA}$-*CYC1-lacZ* reporter containing the UAS from *MEP2* modified to contain additional GATA sequences | *Vijjamarri et al., 2023a* |
| pAK133 | *DCP2-3XHA* cloned in YCplac33 under its native promoter. | This study |

**Table 3.** Primers employed.

| Primer | Sequence (5' to 3') |
|---|---|
| AKV224 | CAAGCCGTTAATGTCGTTATCAATTTCGAT |
| AKV225 | TTCATCTTGTCAGTTGAAATGAATAGTTTA |
| AKV372 | CGACTCTAGAGGATCAAAGAACAATGAACTCTAGAGCATC |
| AKV373 | TCCTGCATAGTCCGGGACGTCATAGGGATAGCCCGCATAGTCAGGAACATCGTATGGGTAAACGGCCGCCTTCCTATGCAAAATGCTTAATAATT |
| AKV374 | TATCCCTATGACGTCCCGGACTATGCAGGATCCTATCCATATGACGTTCCAGATTACGCTCCGGCCGCCTGAAAGAATAAGTGTTATACGTTTTA |
| AKV375 | CGGTACCCGGGGATCAATATCGACAGTTTTAAGAACCGC |

entirety. A complete list of plasmids employed is given in *Table 2*, and all plasmids are available on request. The sequences of primers employed is listed in *Table 3*.

### Cell spotting growth assays

Yeast transformants harboring plasmids containing *EDC3*, *SCD6,* or empty vector were grown to mid-logarithmic phase at 30°C in liquid synthetic complete medium (SC) without uracil (SC-U). Cultures were diluted to $OD_{600}$ of 1.0 and 10-fold serial dilutions were spotted on agar medium of the same composition and incubated at 30°C or 37°C for 2 days.

### Polysome profiling

Polysome profiling was conducted as described previously (*Zeidan et al., 2018*). In brief, strains were cultured in YPD medium at 30°C to mid-logarithmic phase ($OD_{600}$ of ~0.5–0.6). Fifteen $A_{260}$ units of WCEs were resolved on a 10–50% sucrose gradient by centrifugation at 35,000 rpm. Gradients were fractionated with continuous scanning at 260 nm. Areas under the $A_{260}$ tracings of polysome and monosome peaks were calculated using ImageJ software and used to calculate polysome to monosome (P/M) ratios in mutants and WT.

### Ribosome footprint profiling

Duplicate cultures of yeast strains HFY114 (WT W303), SYY2352 (*scd6Δ*), FZY862 (*edc3Δ*), and FZY858 (*scd6Δedc3Δ*) were cultured in YPD medium at 30°C to $OD_{600}$~0.6–0.7. Yeast cells were quickly filtered, frozen in liquid nitrogen and stored at –80C. Cells were lysed in a freezer mill and RPFs were isolated and used for cDNA library preparation as described previously (*Vijjamarri et al., 2023b*). Single-end 100 bp Illumina sequencing was performed by the National Heart, Lung, and Blood Institute (NHLBI) DNA Sequencing and Genomics Core facility (Bethesda, MD).

### RNA sequencing with spike-in normalization and CAGE analysis

The same lysates used for RPF library preparation were used to prepare RNA-Seq libraries after adding ERCC RNA Spike-In Control Mix 1 (Thermo Fisher Scientific, Cat. # 4456740). Total RNA was extracted from cell lysates using QIAzol Lysis reagent (Qiagen Cat. # 79306) and miRNeasy Mini Kit (Qiagen, Cat. 217004). Twenty µg of total RNA of each sample was subjected to RNase-free DNase I (Roche, Cat. # 04716728001) treatment and then processed with the RNA Clean and Concentrator Kit (Zymo, Cat. R1018). An aliquot of 2.4 µl of 1:100 diluted ERCC RNA Spike-In Control Mix 1 was added to 1.2 µg of each RNA sample and submitted to the NHLBI DNA Sequencing and Genomics Core facility (Bethesda, MD) for cDNA library preparation and Illumina sequencing. CAGE sequencing and RNA-Seq were conducted in parallel on two biological replicates of total RNA isolated from the same WT and *scd6Δedc3Δ* strains, cultured under the same conditions as employed for ribosomal profiling, exactly as described previously (*Vijjamarri et al., 2023a*).

### Rpb1 ChIP-Seq with spike-in normalization

Triplicates of wild-type and *scd6Δedc3Δ* yeast strains were cultured in rich YPD medium at 30°C to OD600=0.6–0.8. Chromatin extracts were prepared from formaldehyde cross-linked cells as described previously (*Qiu et al., 2016*). *S. pombe* chromatin was added to each chromatin sample as a spike-in control prior to immunoprecipitation with Rpb1 antibodies, and ChIP-Seq DNA libraries

were prepared as described previously (*Zheng et al., 2023*) subjected to 50 bp paired-end Illumina sequencing by the NHLBI DNA Sequencing and Genomics Core facility (Bethesda, MD) and analyzed as previously described (*Zheng et al., 2023*).

## Western blot analysis

For the results in *Figure 6B–C*, WCEs were prepared by trichloroacetic acid (TCA) extraction as previously described (*Reid and Schatz, 1982*) and immunoblot analysis was conducted as described previously (*Nanda et al., 2009*). After electroblotting to nitrocellulose membranes (Bio-Rad 1620094), membranes were probed with antibodies against Atp20, Cox14, Pet10, Qcr8, Sdh4 (kindly provided by Dr. Nikolaus Pfanner), Cyb2 (kindly provided by Dr. Thomas Fox), Idh1 (Abnova, PAB19472), Cit2 (https://www.antibodies-online.com/, ABIN4889057), and Gcd6 (*Bushman et al., 1993*). Secondary antibodies employed were HRP-conjugated anti-rabbit (Cytiva, NA9340V), anti-mouse IgG (Cytiva, NA931V), and anti-goat IgG (Abnova, PAB29101). Detection was performed using enhanced chemiluminescence (ECL) Western Blotting Detection Reagent (Cytiva, RPN2016) and the Azure 200 gel imaging biosystem. NIH ImageJ was employed to analyze images for quantification. For analysis of Cox2 levels in *Figure 6D*, total proteins were TCA-extracted as described previously (*Rashida et al., 2021*) and subjected to Western blot analysis as recently described (*Niphadkar et al., 2024*) using mouse monoclonal antibody MTCO2 (4B12A5) from Invitrogen.

## TMT-MS analysis of global protein abundance

TMT-MS analysis was conducted as described previously (*Vijjamarri et al., 2023a*) with the following modifications. Replicate cultures of WT, *edc3Δ, scd6Δ,* and *scd6Δedc3Δ* strains were cultured in YPD medium for ~3 doublings to $OD_{600}$ of ~0.6, and harvested by centrifugation for 5 min at 3000×*g*. Cells were resuspended in nuclease-free water, collected by centrifugation, and stored at –80°C. WCEs were prepared in freshly prepared 8 M Urea, 25 mM triethylammonium-bicarbonate (TEAB; Thermo Scientific, 90114) by washing the cell pellets once and resuspending again in the extraction buffer, then vortexing with glass beads in the cold room. Lysates were clarified by centrifugation at 13,000×*g* for 30 min and the quality of extracted proteins was assessed following SDS-PAGE using GelCode Blue Stain (Thermo Scientific, 24592) and quantified with the Pierce BCA Protein Assay Kit (Thermo Scientific, 23225). Lysates were stored at –80°C. Sample preparation, TMT-MS/MS (*Zecha et al., 2019*), and data analysis were performed at the IDeA National Resource for Quantitative Proteomics.

## Co-immunoprecipitation analysis

Transformants of *DHH1-TAP* strains were cultured in 50 ml YPD at 30°C to $OD_{600} \approx 0.6$–0.8, harvested by centrifugation, and washed with 1 ml lysis buffer (LB) containing 0.5 mM PMSF. Cell pellets were resuspended in 500 μl LB supplemented with 0.5 mM PMSF and 1x protease inhibitor cocktail (Roche Cat # 5056489001) and lysed by vortexing in 200 μl glass beads (8 cycles: 30 s on, 45 s off) on ice in a cold room. Lysates were clarified by centrifugation (15,000 rpm, 20 min, 4°C), and protein concentrations determined using the Bradford assay. Approximately 1 μg of lysate was incubated with IgG sepharose beads pre-treated with 5% BSA for 2 h at 4°C. Beads were washed five times with LB containing 0.5 mM DTT. Bound complexes were eluted with 500 μl LB +0.5 mM DTT containing 2 μl TEV protease (1 mg/ml, 10 U/μl; Invitrogen 12575–015) at room temperature for 2 hr. Eluates were precipitated with 10% cold TCA on ice for 10 min, pelleted by centrifugation (13,000 rpm, 10 min, 4°C), washed with cold acetone, air-dried, resuspended in 50 μl 2x SDS PAGE sample buffer, neutralized with 50 μl unadjusted 1 M Tris, and boiled for 5 min. Samples were subjected to immunoblot analysis as described above using rabbit polyclonal TAP antibodies (Thermo Fisher Scientific Cat# CAB1001) and mouse monoclonal antibodies against HA (12CA5) (Roche 10522600) for detecting TAP and HA tags.

## Measuring mitochondrial membrane potential

Precultures were grown in SC-Ura (to select for the *URA3* plasmids) to $OD_{600}$ of ~3.0 and used to inoculate YPD medium at $OD_{600}$ of 0.2. Cells were grown to $OD_{600}$ of ~0.6–0.8, incubated with 500 nM TMRM for 1 hr and subjected to flow cytometry to measure dye fluorescence in individual cells as described previously (*Vijjamarri et al., 2023b*). In control samples, 50 μM FCCP was added

to dissipate the membrane potential and reveal non-specific background fluorescence. Results are presented in arbitrary fluorescence units normalized to $OD_{600}$ of the cultures.

## Analysis of polar metabolites of intermediary metabolism

### Yeast cell culturing

Four replicate 15 mL cultures in YEPD medium were prepared for each strain by inoculating with saturated overnight cultures to $OD_{600}$ of 0.1, culturing with shaking at 30°C, and harvesting at $OD_{600}$ of 0.6 by centrifugation in conical 15 mL tubes for 3 min at 3000 ×g in an Avanti J-HC centrifuge pre-cooled to –10°C. Cell pellets were resuspended in 1.8 mL of ice-cold PBS and centrifuged in 2 mL screw-cap tubes in a refrigerated microfuge for 30 s. Supernatants were decanted, tube rims blotted on tissue paper, and cell pellets frozen in dry ice/ethanol for 5 min and stored at –80°C. Frozen cell pellets were shipped on dry ice to the NYU Metabolomics Core Resource Laboratory, where all subsequent analyses described below were conducted.

### Extraction of metabolites

Frozen cell pellets were thawed on wet ice. Extraction buffer, consisting of 80% methanol (Fisher Scientific) and 500 nM metabolomics amino acid mix standard (Cambridge Isotope Laboratories, Inc), was prepared and placed on dry ice. Samples were extracted by mixing cell pellets with extraction buffer at 10 mg/mL (determined by sample OD measurements) in 2.0 mL screw-cap vials containing ~100 µL of disruption beads (Research Products International, Mount Prospect, IL). Each sample was homogenized for 10 cycles on a bead blaster homogenizer (Benchmark Scientific, Edison, NJ), with each cycle consisting of 30 s homogenization at 6 m/s followed by a 30 s pause. Samples were centrifuged at 21,000 g for 3 min at 4°C. Aliquots of 450 µL were transferred to 1.5 mL tubes, dried in a Speedvac (Thermo Fisher, Waltham, MA), reconstituted in 50 µL of Optima LC/MS grade water (Fisher Scientific, Waltham, MA), sonicated for 2 min, and centrifuged at 21,000 g for 3 min at 4°C. Aliquots of 20 µL were transferred to LC vials containing glass inserts for analysis, and the remaining samples stored at –80°C.

### LC-MS/MS with the hybrid metabolomics method

Samples were subjected to an LC-MS analysis to detect and quantify known peaks. Extraction of polar metabolites was carried out on each sample based on a previously described method (*Jones et al., 2014*). LC was conducted using a Millipore ZIC-pHILIC (2.1×150 mm, 5 µm) column coupled to a Dionex Ultimate 3000 system with gradient elution conducted at 25°C with a flow rate of 100 µL/min using the following buffers (A) 10 mM ammonium carbonate in water, pH 9.0, and (B) neat acetonitrile. The gradient profile was as follows: 80–20% B (0–30 min), 20–80% B (30–31 min), 80–80% B (31–42 min). Injection volume was set to 2 µL for all analyses with a 42 min total run time per injection.

MS was conducted by coupling the LC system to a Thermo Q Exactive HF mass spectrometer operating in heated electrospray ionization mode (HESI). Method duration was 30 min with a polarity switching data-dependent Top five method for both positive and negative modes. Spray voltage for both positive and negative modes was 3.5 kV and capillary temperature was set to 320°C with a sheath gas rate of 35, aux gas of 10, and max spray current of 100 µA. The full MS scan for both polarities utilized 120,000 resolution with an AGC target of $3 \times 10^6$ and a maximum IT of 100 ms, and the scan range was from 67 to 1000 m/z. Tandem MS spectra for both positive and negative mode used a resolution of 15,000, AGC target of $1 \times 10^5$, maximum IT of 50 ms, isolation window of 0.4 m/z, isolation offset of 0.1 m/z, fixed first mass of 50 m/z, and 3-way multiplexed normalized collision energies (nCE) of 10, 35, 80. The minimum AGC target was $1 \times 10^4$ with an intensity threshold of $2 \times 10^5$. All data were acquired in profile mode.

### Relative quantification and statistical analyses of polar metabolites

The resulting Thermo RAW files were converted to SQLite format using an in-house Python script to enable downstream peak detection and quantification. The available MS/MS spectra were first searched against the NIST17 MS/MS (*Simón-Manso et al., 2013*), METLIN (*Smith et al., 2005*), and respective Decoy spectral library databases using an in-house data analysis Python script adapted from our previously described approach for metabolite identification false discovery rate control (FDR)

(*Wang et al., 2018*; *Wang et al., 2020*). Metabolite peaks were extracted based on the theoretical m/z of the expected ion type, e.g., [M+H]$^+$, with a 15 part-per-million (ppm) tolerance and a ± 0.2 min peak apex retention time tolerance within an initial retention time search window of ±0.5 min. For all the group-wise comparisons, t-tests were performed using the Python SciPy (1.5.4) (*Virtanen et al., 2020*) library to test for differences and generate statistics for the downstream analyses. For the pair-wise t-tests, any metabolite with a *p*-value <0.01 was considered significantly regulated (up- or down-) for prioritization in the subsequent analyses.

Coverage of a library of polar metabolites of major pathways of intermediary metabolism allowed detection of 130 of the 147 metabolites examined in at least 4 samples, and 91 detected in all 20 samples after background threshold correction. Instrument performance was assessed using the internal standards added to the samples during metabolite extraction and instrument mass accuracy was within tolerance (–2.0 ppm), LC column performance was stable (0.16 min RT range) and internal standard response variability was 13% across the samples. The resulting data were analyzed by principal components analysis (*Figure 7A*), by constructing heatmaps with unsupervised hierarchical clustering of the imputed matrix values utilizing the R library pheatmap (1.0.12). GraphPad Prism 9 (9.4.1, GraphPad Software, San Diego, CA), and volcano plots (generated utilizing R library script Manhattanly (0.2.0)), in addition to the statistical comparisons summarized in *Figure 7—source data 1*.

### $^{13}C_6$-glucose metabolic flux measurements

Three replicate cultures of each strain were grown in YP medium with 2% unlabeled glucose to OD$_{600}$ of 0.6–0.7 and then shifted to YP with 1% unlabeled glucose and cultured for 20 min. $^{13}C_6$-labeled glucose (1%) was then added and growth was continued for 8 min. Cells were collected, and metabolites were extracted, derivatized, and subjected to mass spectrometry as described previously, with the addition of additional masses for assessing the label incorporation into the specific labeled intermediates that were analyzed (*Walvekar et al., 2018*).

### ATP measurements

Cells were cultured in YPD to OD$_{600}$ of 0.6–0.7 and treated or untreated with 5 mM sodium azide for 30 min at 30°C. Five OD$_{600}$ units of cells were harvested and resuspended in 300 μL ice-cold 5% trichloroacetic acid (TCA), incubated on ice for 15 min, and diluted 50-fold in 20 mM Tris-HCl (pH 7.5)–0.1% TCA, after which 10 μL was mixed with 90 μL of ATP Determination reaction mixture (Cat. # A22066, Thermo Fisher Scientific) and incubated at room temperature for 5 min. Units of Firefly Luciferase were measured in a Berthold Centro XS$^3$ LB 960 Luminometer, and ATP levels were calculated from an ATP standard curve generated using the same assay.

### Additional data visualization and statistical analyses

Notched box-plots were constructed using a web-based tool at http://shiny.chemgrid.org/boxplotr/. In all such plots, the upper and lower boxes contain the second and 3$^{rd}$ quartiles and the band gives the median. If the notches in two plots do not overlap, there is roughly 95% confidence that their medians differ. Density scatter plots, Venn diagrams, significance testing of gene set overlaps in Venn diagrams using the hypergeometric distribution, hierarchical clustering analysis, and gene ontology (GO) analysis were all conducted as described previously (*Vijjamarri et al., 2023a*).

### Acknowledgements

We are grateful to Drew Jones, Tori Rodrick, and the NYU Langone Medical Metabolomics Lab for valuable advice, metabolomics data acquisition and analysis, Samuel Mackintosh and IDeA National Resource for Quantitative Proteomics for TMT-MS analysis, and the NHLBI DNA Sequencing and Genomics Core for all next-generation DNA sequencing. We thank Feng He and Allan Jacobson and Bertrand Séraphin for gifts of yeast strains, Nikolaus Pfanner and Thomas Fox for gifts of antibodies, and Henry Zhang for help with bioinformatics. We are grateful to Jon Lorsch for financial support of DNA sequencing and invaluable advice, and all other members of our laboratories for helpful comments and suggestions. We thank the NHLBI DNA Sequencing and Genomics Core and NIH HPC Biowulf cluster for computational support. This research was supported [in part] by the Intramural Research Program of the National Institutes of Health (NIH). The contributions of the NIH author(s) are considered Works of the United States Government. The findings and conclusions presented in this

paper are those of the author(s) and do not necessarily reflect the views of the NIH or the U.S. Department of Health and Human Services. C.O. and M.L.G. were supported by NIH grants R01 HL117880 and R35 GM149271, and SL acknowledges support from a DBT-Wellcome Trust India Alliance Senior Fellowship IA/S/21/2/505922.

# Additional information

## Competing interests

Alan G Hinnebusch: Reviewing editor, eLife. The other authors declare that no competing interests exist.

## Funding

| Funder | Grant reference number | Author |
|---|---|---|
| National Institutes of Health | HD008899-06 | Rakesh Kumar<br>Fan Zhang<br>Anil Kumar Vijjamarri<br>Alan G Hinnebusch |
| National Institute of General Medical Sciences | R01 HL117880 | Chisom Onu<br>Miriam L Greenberg |
| National Institute of General Medical Sciences | R35 GM149271 | Chisom Onu<br>Miriam L Greenberg |
| Wellcome Trust/DBT India Alliance | Senior Fellowship IA/S/21/2/505922 | Shreyas Niphadkar<br>Sunil Laxman |

The funders had no role in study design, data collection and interpretation, or the decision to submit the work for publication. For the purpose of Open Access, the authors have applied a CC BY public copyright license to any Author Accepted Manuscript version arising from this submission.

## Author contributions

Rakesh Kumar, Fan Zhang, Shreyas Niphadkar, Chisom Onu, Anil Kumar Vijjamarri, Data curation, Formal analysis, Investigation, Visualization, Writing – review and editing; Miriam L Greenberg, Sunil Laxman, Conceptualization, Formal analysis, Supervision, Funding acquisition, Project administration, Writing – review and editing; Alan G Hinnebusch, Conceptualization, Formal analysis, Supervision, Funding acquisition, Visualization, Writing – original draft, Project administration, Writing – review and editing

## Author ORCIDs

Rakesh Kumar ⓘ https://orcid.org/0009-0003-0349-0914
Shreyas Niphadkar ⓘ https://orcid.org/0009-0002-4028-395X
Chisom Onu ⓘ https://orcid.org/0000-0002-3338-5141
Miriam L Greenberg ⓘ https://orcid.org/0000-0001-7724-5426
Alan G Hinnebusch ⓘ https://orcid.org/0000-0002-1627-8395

Reviewer #1 (Public review): https://doi.org/10.7554/eLife.102287.3.sa1
Reviewer #2 (Public review): https://doi.org/10.7554/eLife.102287.3.sa2
Reviewer #3 (Public review): https://doi.org/10.7554/eLife.102287.3.sa3
Author response https://doi.org/10.7554/eLife.102287.3.sa4

# Additional files

## Supplementary files

MDAR checklist

Source data 1. Parallel RNA-Seq and Ribo-Seq analysis of decapping mutants, gene groups dysregulated in decapping mutants or belonging to specific functional categories, and capped/

total mRNA ratios, codon optimality scores, codon protection indices, and Dhh1 occupancy values for all mRNAs. Sheets 1-7 labeled 'mutant vs WT all comp' contain the processed data from Ribo-seq and RNA-seq analysis of two or three biological replicates of each strain listing the fold changes in mRNA, ribosome-protected fragments (RPFs), or translational efficiency (TE) between the relevant mutant vs. WT with the p-values and adjusted p-values (FDRs) assigned by DESeq2 analysis. The data for *dhh1Δ*, *dcp2Δ* (*Zeidan et al., 2018*), *pat1Δ*, and *pat1Δdhh1Δ* mutants (*Vijjamarri et al., 2023a*) were reported previously. Sheet 'Gene groups_mRNA analysis' contains the lists of mRNAs (identified by systematic gene name) that are up-regulated or down-regulated in abundance by ≥1.5-fold at FDR < 0.05 in the *scd6Δ*, *edc3Δ*, or *scd6Δedc3Δ* mutants vs. WT, either including or excluding iESR mRNAs for up-regulated transcripts or rESR mRNAs for down-regulated transcripts, as indicated. It also lists the transcripts belonging to the three sectors of Figs. 1B-C, the iESR and rESR transcripts defined previously (*Gasch et al., 2000*), and all of the non-iESR mRNAs up-regulated by ≥1.5-fold at FDR < 0.05 in *dhh1Δ*, *pat1Δ*, or *pat1Δdhh1Δ* mutants, as well as those equally up-regulated by the *dcp2Δ* mutation. Sheet 'Gene groups_TE analysis' contains the lists of mRNAs that are up-regulated or down-regulated in TE by ≥1.41-fold at FDR <0.10 in the different mutants vs. WT. Sheet 'Gene groups_RPF analysis' contains the lists of mRNAs that are up-regulated or down-regulated in RPFs by ≥1.5-fold at FDR <0.05 in the different mutants vs. WT. Sheet 'Pathway gene groups' lists the genes involved in specific pathways (e.g. Ox. Phos.) analyzed for changes in expression or translation. Sheet 'Codon Opt Scores nonESR mRNAs' lists the tRNA adaptation index (tAI), species-specific tRNA adaptation index (stAI), and the average codon stabilization coefficient (AvgCSC) for all non-ESR yeast genes (*Presnyak et al., 2015*; *Radhakrishnan et al., 2016*). Sheet 'Codon Protection Indices' lists the CPI values for all genes determined previously (*Pelechano et al., 2015*). Sheet 'Capped to Total RNA ratios_s6,e3' lists the ratios of transcript numbers per million reads (TPMs) determined by CAGE sequencing of capped mRNAs (C) to TPMs determined by parallel RNA-seq of the same total RNA samples for 5393 genes, determined for *scd6Δedc3Δ* and WT cells. Sheet 'Capped to Total RNA ratios_dhh1' lists the ratios of transcript numbers per million reads (TPMs) determined by CAGE sequencing of capped mRNAs (C) to TPMs determined by parallel RNA-seq of the same total RNA samples for 5129 genes, determined previously for *dhh1Δ* and WT cells (*Vijjamarri et al., 2023a*). Sheet 'Dhh1 occ enrich scores' lists the relative Dhh1 occupancies (enrichment scores) determined globally for yeast mRNAs by RIP-seq analysis (*Miller et al., 2018*) for the 3686 transcripts for which both Dhh1 enrichment score and Ribo-seq and RNA-seq data from the dhh1Δ vs. WT comparison (*Zeidan et al., 2018*) are available. Sheet 'IGV tracks' lists the results of ribosome profiling of the *scd6Δ*, *edc3Δ*, and *scd6Δedc3Δ* mutants vs. WT for the genes selected for gene-browser depictions.

## Data availability

Ribosome profiling, RNA-Seq, ChIP-Seq and CAGE-Seq data discussed in this publication have been deposited in NCBI's Gene Expression Omnibus and are accessible through GEO Series accession numbers GSE270789 and GSE270790. TMT-MS/MS raw proteomics data for all expressed proteins have been deposited in ProteomeXchange with accession number PXD053307. All other data generated or analyzed during this study are included in the manuscript and supporting files; source data files have been provided for all figures.

The following datasets were generated:

| Author(s) | Year | Dataset title | Dataset URL | Database and Identifier |
|---|---|---|---|---|
| Kumar R, Zhang F, Vijjamarri AK, Hinnebusch AG | 2024 | Decapping activators Edc3 and Scd6 act redundantly with Dhh1 in nutrient-replete cells to post-transcriptionally repress starvation-induced pathways [Ribo and RNA-seq] | https://www.ncbi.nlm.nih.gov/geo/query/acc.cgi?acc=GSE270790 | NCBI Gene Expression Omnibus, GSE270790 |

*Continued*

| Author(s) | Year | Dataset title | Dataset URL | Database and Identifier |
|---|---|---|---|---|
| Kumar R, Zhang F, Vijjamarri AK, Hinnebusch AG | 2024 | Decapping activators Edc3 and Scd6 act redundantly with Dhh1 in nutrient-replete cells to post-transcriptionally repress starvation-induced pathways | https://proteomecentral.proteomexchange.org/cgi/GetDataset?ID=PXD053307 | ProteomeXchange, PXD053307 |
| Kumar R, Zhang F, Vijjamarri AK, Hinnebusch AG | 2024 | Decapping activators Edc3 and Scd6 act redundantly with Dhh1 in nutrient-replete cells to post-transcriptionally repress starvation-induced pathways [ChIP-seq] | https://www.ncbi.nlm.nih.gov/geo/query/acc.cgi?acc=GSE270789 | NCBI Gene Expression Omnibus, GSE270789 |

The following previously published datasets were used:

| Author(s) | Year | Dataset title | Dataset URL | Database and Identifier |
|---|---|---|---|---|
| Vijjamarri AK, Niu X, Zhang F, Qiu H, Gupta N, Lin Z, Hinnebusch AG | 2023 | Decapping factor Dcp2 controls mRNA abundance and translation to adjust metabolism and filamentation to nutrient availability II | https://www.ncbi.nlm.nih.gov/geo/query/acc.cgi?acc=GSE220578 | NCBI Gene Expression Omnibus, GSE220578 |
| Vijjamarri AK, Niu X, Zhang F, Qiu H, Gupta N, Lin Z, Hinnebusch AG | 2023 | Decapping factor Dcp2 controls mRNA abundance and translation to adjust metabolism and filamentation to nutrient availability | https://www.ncbi.nlm.nih.gov/geo/query/acc.cgi?acc=GSE216831 | NCBI Gene Expression Omnibus, GSE216831 |
| Vijjamarri AK, Gupta N, Niu X, Zhang F, Kumar R, Lin Z, Hinnebusch AG | 2023 | mRNA decapping activators Pat1 and Dhh1 regulate transcript abundance and translation to tune cellular responses to nutrient availability | https://www.ncbi.nlm.nih.gov/geo/query/acc.cgi?acc=GSE224774 | NCBI Gene Expression Omnibus, GSE224774 |
| Zhang F, Zeidan Q, Hinnebusch AG | 2019 | Conserved mRNA-granule component Scd6 targets Dhh1 to repress translation initiation and activates Dcp2-mediated mRNA decay in vivo | https://www.ncbi.nlm.nih.gov/geo/query/acc.cgi?acc=GSE114892 | NCBI Gene Expression Omnibus, GSE114892 |

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
