## [Editor Report · eLife Assessment]

This **important** study reports on the redundant roles of the decapping activators Edc3 and Scd6 in orchestrating post-transcriptional programs to modulate metabolic responses to nutrients in yeast. The authors employed mutagenesis studies in conjunction with a battery of transcriptome-wide analyses to provide **convincing** evidence supporting their conclusions. Considering the broad implications of post-transcriptional regulation of gene expression, this study will be of interest across a variety of biomedical disciplines ranging from biochemistry and molecular and cellular biology to those specializing in studying various pathologies.

---

## [Referee Report · Reviewer #1 (Public review)]

Summary:

mRNA decapping and decay factors play critical roles in post-transcriptionally regulating gene expression. Here, Kumar and colleagues investigate how deleting two yeast decapping enhancer proteins (Edc3 and Scd6), either alone or in tandem, affects the transcriptome. Using RNA-Seq, CAGE-Seq and ribosome profiling, they conclude that these factors generally act in a redundant fashion, with a mutant lacking both proteins showing an increased abundance of select mRNAs. As these upregulated transcripts are also upregulated in mutants lacking the decapping enzyme, Dcp2, and show no increases in transcription of their cognate genes, the authors conclude that this is at the level of mRNA decapping and decay. This was further supported by CAGE-Seq analyses carried out in WT cells and the scd∆6edc3∆ double mutant. Their ribosome profiling data also lead them to conclude that Scd6 and Edc3 display functional redundancy and cooperativity with Dhh1/Pat1 in repressing the translation of specific transcripts. Finally, as their data suggest that Scd6 and Edc3 repress mRNAs coding for proteins involved in cellular respiration, as well as proteins involved in the catabolism of alternative carbon sources, they go on to show that these decapping activators play a role in repressing oxidative phosphorylation.

Strengths:

Overall, this manuscript is well-written and contains a large amount of compelling high-quality data and analyses. At its core, it helps to shed light on the overlapping roles Edc3 and Scd6 have in sculpting the yeast transcriptome.

Weaknesses:

While not essential, it would be interesting if the authors carried out add-back experiments to determine which domain within Scd6/Edce3 plays a critical role for enforcing the regulation that they see? Their double mutant now puts them in a perfect position to carry out such experiments.

---

## [Referee Report · Reviewer #2 (Public review)]

Summary:

This manuscript by Kumar and Zhang presents compelling evidence that Edc3 and Scd6 decapping activators, present a high degree of redundancy that can only be overcome by double mutants of both. In addition, the authors provide strong evidence for their role in regulating starvation-induced pathways as evidenced by measurements of mitochondrial membrane potential, metabolomics and analysis of the flux of Krebs cycle intermediates.

Strengths:

Kumar, Zhang et al provide multiple source of evidence of the direct mechanism of Edc3 and Scd6, by using and comparing different approaches such as mRNA-seq, ribosome occupancies and translational efficiencies. By extensive analysis the authors show that this complex can also regulate genes outside the Environmental Stress Response (non-iESR) that are significantly up-regulated in all three mutants. Remarkably, the gene ontology analysis of these non-iESR genes identify enrichment for mitochondrial proteins that are implicated in the Krebs cycle. Overall, this study adds novel mechanistic insight into how nutrients control gene expression by modulating decapping and translational repression.

Weaknesses:

The authors show very nicely that growth phenotypes from scd6Δedc3∆ can be rescued by transformation of EDC3 (pLfz614-7) or SCD6 (pLfz615-5). Future work could make use of these rescue strategies, for example as a platform to further characterise protein-protein interactions between Edc3, Scd6 and Dhh1.

---

## [Referee Report · Reviewer #3 (Public review)]

Summary:

In this paper, Kumar et al investigated the role of two decapping activators, Edc3 and Scd6, in regulating mRNA decay and translation in yeast. Using a variety of approaches including RNA-seq, ribosome profiling, proteomics, polysome analysis, and metabolomics the authors demonstrate that whereas single deletions of Edc3 or Scd6 have modest effects, the double mutant leads to increased abundance of mRNAs, many of which overlap with those targeted by the decapping activators Dhh1 and Pat1. The data suggest that Edc3 and Scd6 function redundantly to recruit Dhh1 to the Dcp2 decapping complex, thereby promoting mRNA turnover and translational repression. The authors show that these factors cooperate with Dhh1/Pat1 to repress transcripts involved in respiration, mitochondrial function, and alternative carbon source utilization, linking post-transcriptional regulation to nutrient responses. The study establishes Edc3 and Scd6 as important, but redundant regulators that fine-tune gene expression and metabolic adaptation in response to nutrient availability.

Strengths:

The paper has several strengths, including the comprehensive approach taken by the authors using multiple experimental techniques (RNA-seq, ribosome profiling, Western blotting, TMT-MS, polysome profiling, and metabolomics) to provide multiple lines of evidence to support their conclusions. The authors demonstrate clear redundancy of the factors by using single and double mutants for Edc3 and Scd6 and their global approach enables an understanding of these factors' roles across the yeast transcriptome. The work connects post-transcriptional processes to nutrient-dependent gene regulation, providing insights into how cells adapt to changes in their environment. The authors demonstrate the redundant roles of Edc3 and Scd6 in mRNA decapping and translation repression. Their RNA-seq and ribosome profiling results convincingly show that many mRNAs are derepressed only in the double mutants, confirming their hypothesis of redundancy. Furthermore, the functional cooperation between Edc3/Scd6 and Dhh1/Pat1 in regulating specific metabolic pathways, including mitochondrial function and carbon source utilization, is supported by the metabolomic data.

Weaknesses:

The study uses indirect evidence to support claims about the effect on mRNA stability rather than directly measuring mRNA stability. However, the combination of Pol II occupancy and RNA abundance measurements is consistent with the claims regarding mRNA stability. The addition of new experiments in the revision co-IPing Dhh1 and Dcp2 strengthens the argument that Edc3 and Scd6 recruit these factors.

---

## [Author Response]

The following is the authors’ response to the original reviews.

**Reviewer #1 (Public review):**
Strengths:Overall, this manuscript is well-written and contains a large amount of high-quality data and analyses. At its core, it helps to shed light on the overlapping roles of Edc3 and Scd6 in sculpting the yeast transcriptome.Weaknesses:(1) While the data presented makes conclusions about mRNA stability based on corresponding ChIP-Seq analyses and analyzing other mutants (e.g. Dcp2 knockout), at no point is mRNA stability actually ever directly assessed. This direct assessment, even for select transcripts, would further strengthen their conclusions.

We appreciate the reviewer’s concern but wish to emphasize that we conducted ChIP-Seq analysis of RNA Polymerase II occupancies in the CDSs of all genes, known to be a reliable indicator of transcription rate, and found only small increases in Pol II occupancies that cannot account for the increased transcript levels of the cohort of mRNAs up-regulated in the scd∆6edc3∆ double mutant (Fig. 3E). This provides strong evidence that increased transcription is not the main driver of increased mRNA abundance in this mutant. Bolstering this conclusion, we showed that the Hap2/Hap3/Hap4/Hap5 complex of transcription factors responsible for induction of Ox. Phos. genes was not activated in scd6Δedc3Δ cells in glucose medium (Fig. 6F(ii)); nor was the Adr1 activator of CCR genes activated (Fig. S9C(i)), ruling out transcriptional induction of their target genes in glucose-replete scd6Δ/edc3Δ cells and instead favoring reduced degradation as the mechanism underlying derepression of Ox. Phos. and CCR gene transcripts in this mutant. In Fig. 3B, we further showed that the majority of mRNAs up-regulated in the scd6Δedc3Δ double mutant are also derepressed by dcp2Δ, and in Fig. 3D that the mRNAs up-regulated in scd∆6edc3∆ cells exhibit a higher than average codon protection index (CPI) indicating a heightened involvement of decapping and co-translational degradation by Xrn1 in their decay. To provide additional support for our conclusion, we have conducted new experiments to measure the abundance of capped mRNAs genome-wide by CAGE sequencing of total mRNA in both WT and scd∆6edc3∆ cells. As established previously, normalizing CAGE TPMs to total mRNA TPMs determined by RNA-Seq, dubbed the C/T ratio, provides a reliable measure of the capped proportion of each transcript. The new data presented in Fig. 3C indicate that the mRNAs up-regulated in the scd∆6edc3∆ mutant have significantly lower than average C/T ratios in WT cells, whereas the C/T ratios for the down-regulated transcripts are higher than average, and that these differences between the two groups and all expressed mRNAs are diminished in the scd∆6edc3∆ double mutant. These are the results expected if the up-regulated mRNAs are selectively targeted for decapping in WT cells dependent on Edc3/Scd6, whereas the downregulated mRNAs are targeted by Edc3/Scd6 less than the average transcript. In the original version of the paper, we came to the same conclusion by analyzing our previous CAGE data for the dhh1∆ mutant for the same transcripts dysregulated scd∆6edc3∆ cells, now presented as supportive data in Fig. S3F. Finally, we added the fact that among all four Dhh1 target mRNAs examined in the previous study of He et al. (2022) and found here to be up-regulated selectively in the scd6∆edc3∆ double mutant (Fig. S10), two of them (SDS23 and HXT6) were shown directly to have longer half-lives in dhh1∆ vs. WT cells by He et al. (2018). Hence, the combined evidence is compelling that selective up-regulation of particular mRNAs in the scd∆6edc3∆ mutant results from diminished decapping/decay rather than enhanced transcription; and we feel that the additional supporting evidence that would be provided by measuring half-lives of a small group of up-regulated transcripts would not justify the considerable effort required to do so. Moreover, the standard approach for such experiments of impairing transcription with an inhibitor of Pol II or a Pol II Ts^-^ mutation has been criticized because of the known buffering (suppression) of mRNA decay rates in response to impaired transcription.

(2) Scd6 and Edc3 show a high level of functional redundancy, as demonstrated by the double mutant. As these proteins form complexes with other decapping factors/activators, I'm curious if depleting both proteins in the double mutant destabilizes any of these other factors. Have the authors ever assessed the levels of other key decapping factors in the double mutants (i.e. Dhh1, Pat1, Dcp2...etc)? I wonder if depleting both proteins leads to a general destabilization of key complexes. It would also be interesting to see if depleting Edc3 or Scd6 leads to a concomitant increase in the other protein as a compensatory mechanism.

We thank the reviewer for this insight. Examining our Ribo-Seq and TMT-MS data revealed that Dhh1 expression and steady-state abundance are increased ~2-fold in the scd6∆edc3∆ strain, indicating that the up-regulation of many of the same mRNAs by scd6∆edc3∆ and dhh1∆ does not result indirectly from reduced levels of Dhh1 in the scd6∆edc3∆ mutant. The predicted increased in Dhh1 expression might signify a compensatory response to the absence of Scd6/Edc3. We also observed an ~40% reduction in Dcp2 translation (RPFs) and mRNA abundance in the scd6∆edc3∆ strain, which might contribute to the up-regulation of mRNAs dysregulated in this mutant. However, our new immunoblot analyses revealed no significant reduction in steady-state Dcp2 levels in scd6∆edc3∆ cells (Input lanes in Figs. 3F and S4C(i)-(ii)). Moreover, our previous finding that the majority of mRNAs subject to NMD, up-regulated by both upf1∆ and dcp2∆, are not upregulated by scd6∆edc3∆ implies that Dcp2 abundance in scd6∆edc3∆ cells is adequate for normal levels of NMD and favors a direct role for Scd6/Edc3 in accelerating degradation of most transcripts up-regulated in this mutant. We have added these points to the DISCUSSION.

(3) While not essential, it would be interesting if the authors carried out add-back experiments to determine which domain within Scd6/Edce3 plays a critical role in enforcing the regulation that they see. Their double mutant now puts them in a perfect position to carry out such experiments.

We agree with the reviewer that our scd6∆edc3∆ strain provides an opportunity to dissect the Scd6 and Edc3 proteins to determine which domains and motifs of each protein are most critically required for their functions in activating mRNA decay. However, if conducted thoroughly, this would entail an extensive analysis requiring a combination of genetics, biochemistry and genomics. Considering the large amount of data already presented in 43 and 34 panels of main and supplementary figures, respectively, we feel that these additional experiments would be conducted more appropriately as a stand-alone follow-up study.

**Reviewer #2 (Public review):**
Weaknesses:The authors show very nicely in Figure S1A that growth phenotypes from scd6Δedc3∆ can be rescued by transformation of EDC3 (pLfz614-7) or SCD6 (pLfz615-5). The manuscript might benefit from using these rescue strategies in the analysis performed (e.g. RNA-seq, ribosome occupancies, and translational efficiencies). Also, these rescue assays could provide a good platform to further characterise the protein-protein interactions between Edc3, Scd6, and Dhh1.

We responded to this point immediately above in responding to Rev. #1.

**Reviewer #3 (Public review):**
Weaknesses:The limitations of the study include the use of indirect evidence to support claims that Edc3 and Scd6 recruit Dhh1 to the Dcp2 complex, which is inferred from correlations in mRNA abundance and ribosome profiling data rather than direct biochemical evidence.

While the reviewer makes a valid point, it is important to note that the greater correlations between effects of scd6∆edc3∆ with those conferred by dhh1∆ vs. pat1∆ also extended to changes in metabolites (Fig. 7A-C). To provide more direct evidence that Edc3 and Scd6 recruit Dhh1 to the Dcp2 complex, we have now conducted co-immunoprecipitation experiments (presented in new Figs. 3F and S5) demonstrating that association of Dhh1 with Dcp2 is diminished in the scd6∆edc3∆ double mutant but not in either scd6∆ or edc3∆ single mutant, thus providing biochemical support for our proposal.

Also, there is limited exploration of other signals as the study is focused on glucose availability, and it is unclear whether the findings would apply broadly across different environmental stresses or metabolic pathways. Nonetheless, the study provides new insights into how mRNA decapping and degradation are tightly linked to metabolic regulation and nutrient responses in yeast. The RNA-seq and ribosome profiling datasets are valuable resources for the scientific community, providing quantitative information on the role of decapping activators in mRNA stability and translation control.

While not disputing the facts of this comment, we think it is unjustified to label as a weakness that our study focused on glucose-grown cells considering the large amount of new data and insights made possible by our multi-omics approach, presented in >70 separate figure panels and nine supplementary datafiles, which the reviewer has characterized as being valuable to the scientific community. Parallel studies in non-preferred carbon or nitrogen sources are underway and represent large-scale investigations in their own right, for which the current dataset in glucose-replete cells provides the critical reference condition.

**Reviewer #1 (Recommendations for the authors):**
The authors made a note that a set of 37 mRNAs is repressed exclusively by Edc3 with little contribution by Scd6, a list that includes the RPS28B mRNA. Edc3 has been previously reported to promote the decay of this mRNA in a deadenylation-independent fashion by binding to an element in its 3'UTR (PMIDs 15225544, 24492965). Can the authors comment on whether Edc3 may be binding to similar elements in the 3'UTRs of these transcripts in their shortlist? This could be an interesting topic matter for discussion as well.

While an interesting idea, this seems unlikely because the 3’UTR sequence in RPS28B mRNA was shown to bind Rps28 protein itself to confer heightened decapping and decay dependent on Edc3 in a negative autoregulatory loop that exerts tight control over Rps28 protein levels. It would be surprising if Edc3mediated repression of the other 36 mRNAs would involve Rps28 as none of them encode cytoplasmic ribosomal proteins. Nevertheless, we searched for a conserved motif among the 3’UTRs of the 37 mRNAs using the MEME suite and found enrichment for motifs identified for RNA binding proteins Hrp1 and Nab2 and two novel motifs, but none of these motifs could be recognized within in the Rps28 autoregulatory loop. We have chosen not to comment on these findings in the revised manuscript to avoid lengthening it unnecessarily with inconclusive observations.

**Reviewer #2 (Recommendations for the authors):**
The authors show very nicely in Figure S1A that growth phenotypes from scd6Δedc3∆ can be rescued by the transformation of EDC3 (pLfz614-7) or SCD6 (pLfz615-5). The manuscript might benefit from using these rescue strategies on the analysis performed (e.g. RNA-seq, ribosome occupancies, and translational efficiencies); or expressing truncated mutants of EDC3 (pLfz614-7) or SCD6 (pLfz615-5), to show that they can act as dominant negative competitors, either on the binding to Dhh1 and Dcp2.

We addressed this comment above in our response to this Reviewer.

**Reviewer #3 (Recommendations for the authors):**
(1) Labels such as "mRNA_up_s6,e3" are not defined in figures or the text. I suggest clearer sample labeling throughout.

The labels had been defined at first mention in the RESULTS but are now indicated there more explicitly, as well as in the legend to Fig. 1.

(2) In Figure 1D it is surprising that the mRNA profile has a peak in the 5' UTR. I would expect to see such a peak in ribosome footprinting data. Is it possible these are incorrectly labeled?

The figure is correctly labeled. Generally, one does not expect to see RPFs in the 5’UTR region unless there is an efficiently translated uORF, which appears not to be the case for MDH2.

In general, the information in this panel and C is inadequate. None of the numbers are clearly explained in the figure legend or in the figure.

We had cited the legend to Fig. S3C for details of all such gene browser images but have now inserted this information into the Fig. 1D legend, at the first occurrence of such data in the regular figures.

(3) Figures 1C and 1D are in the wrong order.

Corrected.

(4) Figure 2D is a very complicated Venn Diagram. I suggest using UpSet plots as an alternative to Venn diagrams to more clearly convey overlaps between sets.

We provided additional explanatory text in the Fig. 2D legend to facilitate understanding.

(5) The use of the same color scheme to represent different sets in panels of the same figure is a source of confusion. E.g. the cyan in Figures 2A, 2D, and 2E indicates unrelated categories, but one would think they are related.

The use of the same cyan color in these three figure panels actually does designate results for the same set of 591 mRNAs up-regulated in the three mutants. The application of the color schemes is now mentioned explicitly in Figs. 1, 2, and S3.

(6) Reporting of p-values = 0 in figures is not useful.

Corrected.

(7) The whole manuscript is extremely long which reduces the overall impact. For example, the introduction is six pages long. I suggest reducing redundant text and being more concise to enhance readability.

We tried to streamline the text wherever possible, in particular shortening the Introduction by two pages.

(8) Many abbreviations are used throughout the text that are not introduced the first time they are used.

Corrected throughout.

(9) The ERCC normalization is unclear. Were the spike-ins added before cell lysis to allow estimation of per-cell RNA counts or to the extracted RNA? If added to extracted RNA rather than cells it is not clear to me how the claim can be made regarding increased mRNA abundance in the mutants.

We thank the reviewer for this comment. As we explained in the Methods, 2.4 µl of 1:100 diluted ERCC RNA Spike-In Control Mix 1 was added to 1.2 µg of each total RNA sample prior to cDNA library preparation. Because the majority of total mRNA is comprised of rRNA, this normalization yields the abundance of each mRNA relative to rRNA. Owing to repression of rESR mRNAs encoding ribosomal proteins and biogenesis factors in the scd6∆edc3∆ strain (Fig. S3D), the ribosome content per cell is expected to be reduced in this mutant vs. WT. We showed previously that the isogenic dcp2∆ mutant that elicits an ESR response of similar magnitude, showed a 30% reduction in bulk ribosomal subunits per cell compared to same WT strain examined here {Vijjamarri, 2023 #7866}. Assuming a similar reduction in ribosome abundance in the scd6∆edc3∆ mutant, the changes in mRNA per cell conferred by the scd6∆edc3∆ mutation are expected to be 0.7-fold of the ERCCnormalized values given in Fig. 3E, yielding fold-changes of 2.00 and 0.62 for the mRNA_up and mRNA_dn, groups, respectively, which still differ substantially from the corresponding changes in normalized Rpb1 occupancies of 1.2 and 0.93, respectively. We have added this new analysis to the text of RESULTS.

(10) The use of the terms "up-regulated" and "derepressed" throughout is confusing. Both refer to observed increased abundance of mRNAs, but they imply different causes which are never clearly defined.

We changed all occurrences of “derepressed” to “up-regulated”.